# BANDITS WITH SINGLE-PEAKED PREFERENCES AND LIMITED RESOURCES

**Omer Ben-Porat, Gur Keinan & Rotem Torkan**
Technion—Israel Institute of Technology
Haifa, Israel
omerbp@technion.ac.il, {gur.keinan, torkan.rotem}@campus.technion.ac.il

## ABSTRACT

We study an online stochastic matching problem in which an algorithm sequentially matches $U$ users to $K$ arms, aiming to maximize cumulative reward over $T$ rounds under budget constraints. Without structural assumptions, computing the optimal matching is NP-hard, making online learning computationally infeasible. To overcome this barrier, we focus on *single-peaked preferences*—a well-established structure in social choice theory, where users' preferences are unimodal with respect to a common order over arms. We devise an efficient algorithm for the offline budgeted matching problem, and leverage it into an efficient online algorithm with a regret of $\tilde{O}(UKT^{2/3})$. Our approach relies on a novel PQ tree-based order approximation method. If the single-peaked structure is known, we develop an efficient UCB-like algorithm that achieves a regret bound of $\tilde{O}(U\sqrt{TK})$.

## 1 INTRODUCTION

Modern recommendation systems often face the challenge of personalization at scale—learning individual user preferences while simultaneously satisfying global resource allocation constraints. To illustrate, consider a content platform that must decide which content creators to commission daily, where each creator has a different cost and produces ephemeral content on specific topics. Each user has preferences over all creators' content styles and topics. After commissioning a subset of creators that fit the platform's budget, it matches each user to content from one of these creators, where the same creator's content can be recommended to multiple users. The challenge lies in learning individual user preferences for each creator's content while selecting which creators to commission and how to assign their content to maximize user satisfaction.

This problem fits the combinatorial multi-armed bandit framework, where the decision-maker must choose structured action sets (Chen et al., 2013), such as assigning each user to an item. The goal is to maximize cumulative reward, or equivalently, minimize regret by balancing exploration and exploitation. Unfortunately, combinatorial problems like the one in the example above are NP-complete even for offline settings. Therefore, traditional approaches settle for weaker notions of $\alpha$-regret (Chen et al., 2013), competing against the best efficient computable solution rather than the optimal one. This compromise can be unsatisfying, especially when the solution space is highly structured or the stakes of poor decisions are high.

In this paper, we circumvent this computational barrier by focusing on *single-peaked* (hereinafter SP) preferences—a structure that has been extensively studied in social choice theory since the seminal work of Black (1948). Specifically, it implies that there is an order of the arms such that each user's utility is unimodal. SP preferences appear naturally in numerous domains: voters' preferences over political candidates along an ideological spectrum, consumers' preferences for products varying in a single attribute, and users' preferences for content recommendations based on genre similarity. The single-peaked property has proven helpful in circumventing impossibility results in voting theory (Arrow, 1950) and transforming NP-hard problems into polynomial-time solvable ones (Faliszewski et al., 2009; Elkind & Lackner, 2016).

## 1.1 OUR CONTRIBUTIONS

We study a stochastic combinatorial bandit problem over $T$ rounds with $K$ arms and $U$ users. Each arm $k$ has a cost $c_k$, and the learner may select any subset of arms within a budget $B$, then assign users to the selected arms. Each user derives a stochastic satisfaction from each arm, and the goal is to maximize total user satisfaction. As the general case is NP-complete, we focus on SP preferences (see Definition 2), which enable efficient solutions. Our main contributions are as follows.

**SP instances are statistically challenging** To demonstrate that SP preferences do not trivialize the learning problem, we show that the worst-case regret remains unchanged compared to general preferences. Specifically, we prove that even when the SP order is known, a regret of $\Omega(U\sqrt{TK})$, which matches the lower bound for general preferences, is unavoidable. Furthermore, even if the user peaks are known as well, every algorithm incurs a regret of $\Omega(\max\{U\sqrt{T}, \sqrt{TK}\})$.

**Efficient offline algorithm.** We develop SP-MATCHING, an efficient algorithm that optimally solves the budget-constrained matching problem in $O(K^2 B + K^2 U)$ time when preferences are SP.

**Learning with known SP structure** We consider the case of known SP order and user peaks (but not the cardinal preferences). Such an assumption is justified if, for example, preferences rely on a known ideological spectrum where user peaks correspond to stated or inferred inclinations. We design an efficient UCB-based algorithm, termed MvM, that achieves $\tilde{O}(U\sqrt{TK})$ regret. The key property used in our approach is the existence of a *maximal preferences matrix* in the confidence set of all plausible SP matrices—namely, a matrix that element-wise dominates all others in the set. We use this maximal matrix as the basis for optimistic matching using our offline algorithm.

**Learning with unknown SP structure** We also consider the more challenging case, where the SP structure is unknown. We introduce an explore-then-commit algorithm, named EMC, that devotes an initial exploration phase to collecting reward estimates. Then, a novel PQ tree-based procedure (Booth & Lueker, 1976), which we refer to as EXTRACT-ORDER, recovers an approximate SP order from these estimates. Given this order, the reward estimates are projected onto a nearby SP matrix, and the corresponding offline matching is solved using SP-MATCHING; the resulting matching is committed to for the remainder of the rounds. A careful approximation and concentration analysis yields a regret bound of $\tilde{O}(UKT^{2/3})$, while all computational steps run in $\text{poly}(U, K, B)$ time.

Additionally, due to space constraints, we defer two extensions of our main results to Appendix A: an extension of MvM for *separated instances*, and an extension of EMC to *non-SP instances*.

## 1.2 RELATED WORK

Our research intersects the combinatorial and matching threads of the multi-armed bandits literature, as well as single-peaked preferences from social choice theory.

**Multi-armed bandits.** The multi-armed bandit framework provides the foundation for sequential decision-making under uncertainty (Bubeck et al., 2012; Slivkins, 2019; Lattimore & Szepesvári, 2020). This setting has been extended to the combinatorial domain, where actions are solutions of a combinatorial optimization problem (Combes et al., 2015; Kveton et al., 2015b). While efficient learning algorithms exist for certain combinatorial structures (Neu & Bartók, 2016), many practical instances involve computationally intractable optimization problems. To address this computational barrier, prior work introduced the notion of $\alpha$-regret (Chen et al., 2013), which compares the learner's performance against the best efficiently computable solution rather than the true optimum. This approach has led to the development of general frameworks for achieving $\alpha$-regret guarantees in computationally hard settings (Rizk et al., 2022; Nie et al., 2023). Notably, the single-peaked structure we impose enables us to circumvent computational intractability entirely, allowing us to achieve standard regret bounds rather than settle for $\alpha$-regret. Our setting further incorporates per-round budget constraints, which align with the cardinality and knapsack-type restrictions studied in prior works (Nie et al., 2022; 2023). A closely related line of work considers bandit learning for matching problems (Das & Kamenica, 2005; Liu et al., 2020; Kong et al., 2022; 2024). Another

perspective arises in content recommendation, which can be naturally viewed as a matching problem between users and creators. This has motivated a line of research on strategic content providers (Mladenov et al., 2020; Hron et al., 2023; Immorlica et al., 2024). Particularly, Ben-Porat & Torkan (2023) study a variant where arms become unavailable if they are not selected sufficiently often, and their proposed algorithm has runtime exponential in $K$. In contrast, in our setting, arms remain continuously available (subject to budget constraints), and we devise efficient algorithms.

**Single-peaked preferences.** The study of single-peaked preferences dates back to seminal work in social choice theory (Black, 1948; Arrow, 1950). From a computational perspective, several problems have been investigated, including recognizing whether a given preferences profile admits a single-peaked order and constructing such an order when it exists (Bartholdi III & Trick, 1986; Escoffier et al., 2008). While recognition and construction of single-peaked orders can be solved efficiently, the problem becomes computationally intractable when preferences are nearly but not perfectly single-peaked. Specifically, finding an order that minimizes various distance measures from single-peakedness has been shown to be NP-hard (Faliszewski et al., 2011; Elkind & Lackner, 2014; Escoffier et al., 2021). In Section 6, we contribute to this literature by developing a technique to extract single-peaked orders from approximately single-peaked cardinal preferences.

The single-peaked assumption has profound implications across different domains. In social choice theory, it circumvents classical impossibility results and enables positive outcomes that fail under general preferences. Most notably, Black's median voter theorem (Black, 1948) guarantees the existence of a Condorcet winner under single-peaked preferences, resolving a fundamental challenge in voting theory. From an algorithmic perspective, single-peakedness has been shown to simplify computational challenges: numerous voting problems that are NP-hard under general preferences become polynomial-time solvable when restricted to single-peaked structures (Faliszewski et al., 2009; Elkind & Lackner, 2016). Our work reveals a similar phenomenon in an entirely distinct setting. We demonstrate that in our cardinal matching model with budget constraints, the single-peaked structure enables efficient optimization for problems that are computationally intractable under general preferences.

## 2 PROBLEM DEFINITION

In this section, we formally introduce our model and notation (Section 2.1), address the case of general preferences (Section 2.2), and focus on single-peaked preferences (Section 2.3).

### 2.1 MODEL

We formalize the **C**onstrained **B**andit **R**ecommendation problem (CBR for brevity) as follows. An instance of CBR consists of the tuple $(T, U, K, (D_{u,k})_{u,k}, (c_k)_k, B)$, whose components are described below. The problem involves $U$ users indexed by $[U] = \{1, \ldots, U\}$ and $K$ items (arms) indexed by $[K] = \{1, \ldots, K\}$. The satisfaction of each user $u$ with an item $k$ is stochastic and stationary over time, drawn from a distribution $D_{u,k}$ supported on $[0, 1]$ with expected value $\theta_{u,k}$. We denote the expected reward matrix $\Theta \in [0, 1]^{U \times K}$ by setting its $(u, k)$-th entry to $\theta_{u,k}$. The distributions are assumed to be independent across users and arms.

The goal of the learner is to recommend arms to users so as to maximize their overall satisfaction. Interaction proceeds over $T$ rounds. In each round $t \in [T]$, the learner chooses a *matching* $\pi_t : [U] \to [K]$, namely a function assigning each user a single item. The expected value of the matching is the total expected satisfaction of the users with their assigned arms, $V(\pi_t; \Theta) = \sum_{u \in [U]} \theta_{u,\pi_t(u)}$.

However, not all matchings are feasible. The algorithm has a budget constraint on the selected arms: each selected arm (i.e., there is at least one user $u$ such that $\pi_t(u) = k$) has a cost of $c_k$, and the total cost of the selected arms in every round should remain below a known budget $B \in \mathbb{N}$. Formally, for a matching $\pi$, let $\mathrm{Im}(\pi) = \{k \in [K] : \exists u \in [U] \text{ s.t. } \pi(u) = k\}$. Hence, a matching is feasible only if $\sum_{k \in \mathrm{Im}(\pi_t)} c_k \leq B$. We denote the set of all feasible matchings as $\Pi$. Without loss of generality, we assume that $c_k \leq B$ for all $k \in [K]$, since otherwise such arms could never be selected. As in the motivating example of Section 1, the platform pays the content creators (arms) per creation, so multiple users can be matched to the same arm without incurring additional cost.

At the beginning of each round $t$, the learner has access to the history of past actions and observed rewards, denoted by $H_t$. Upon selecting a matching $\pi_t$, a random reward $r^t_{u,\pi_t(u)} \sim D_{u,\pi_t(u)}$ is generated for each pair $(u, \pi_t(u))$. We assume semi-bandit feedback, meaning the learner observes $r^t_{u,\pi_t(u)}$ for every matched pair $(u, \pi_t(u))$. The performance of the learner $\mathcal{A}$ on a CBR instance $\mathcal{I}$ is measured using the standard notion of regret,

$$R_T(\mathcal{A}, \mathcal{I}) = T \cdot \max_{\pi \in \Pi} V(\pi; \Theta) - \mathbb{E}\left[\sum_{t=1}^{T} \sum_{u \in [U]} r^t_{u,\pi_t(u)}\right],$$

where the expectation is over the algorithm's randomness and the realized rewards. As is standard in bandit settings, we assume $T \geq K, U$. As a useful intermediate step, we also consider the offline version of the problem, where the algorithm is given the expected reward matrix $\Theta$ and must compute the optimal feasible matching, i.e., $\arg\max_{\pi \in \Pi} V(\pi; \Theta)$.

## 2.2 WARM UP: GENERAL PREFERENCES

Without structural assumptions on $\Theta$, achieving low regret with polynomial time algorithms is hopeless. In particular, we establish the following hardness result based on a reduction from the maximum coverage problem (Feige, 1998):

**Theorem 1.** *It is NP-hard to approximate $\arg\max\limits_{\pi \in \Pi} V(\pi; \Theta)$ within any factor better than $(1 - 1/e)$.*

Consequently, since an efficient online algorithm with sublinear regret would imply an efficient approximation algorithm for the offline problem, such an online algorithm cannot exist. We defer the full analysis to Appendix B. If we disregard computational constraints, optimal regret bounds can be attained by, e.g., applying the CUCB algorithm (Chen et al., 2013), assuming access to an exact optimization oracle. Formally,

**Corollary 2.** *There exists a UCB-based algorithm that achieves a regret of $O(U\sqrt{KT \log T})$, which is optimal up to logarithmic factors.*

While this algorithm is statistically optimal, it assumes an optimization oracle that must solve an NP-hard problem at every time step, rendering it computationally impractical.

The literature typically addresses cases where the offline benchmark is intractable using the notion of $\alpha$-regret (Chen et al., 2013; Nie et al., 2022; 2023). This metric uses the best efficiently computable approximation as the baseline, rather than the true optimum. Although it is computationally feasible to achieve sublinear $\alpha$-regret in our setting (e.g., with $\alpha = 0.5$ using greedy oracle), this is a weaker guarantee compared to the standard regret. This limitation motivates our assumption of structural properties for $\Theta$, which enables efficient optimization without sacrificing regret guarantees.

## 2.3 SINGLE-PEAKED PREFERENCES

Single-peaked preferences arise naturally in numerous applications and, as we show, this structure enables polynomial-time algorithms to achieve sublinear regret. We define it formally as follows.

**Definition 1** (PSP matrix). *We say that a matrix $\Theta$ is perfectly single-peaked (PSP for brevity) if for every user $u \in [U]$, there exists an index $p(u; \Theta) \in [K]$ such that*

$$\theta_{u,1} \leq \cdots \leq \theta_{u,p(u;\Theta)} \geq \cdots \geq \theta_{u,K}. \tag{1}$$

In other words, the entries of every row are unimodal. The index $p(u; \Theta)$ is called the *peak* of user $u$, or simply $p(u)$ when $\Theta$ is clear from context; if multiple indices attain the maximum, we fix an arbitrary choice.[1] More generally,

**Definition 2** (SP matrix and SP order). *We say that a matrix $\Theta$ is single-peaked (or SP) if there exists a (total) order $\prec$ on the arms such that the matrix obtained by permuting the columns of $\Theta$ according to $\prec$ is PSP. In such a case, we say that $\prec$ is an SP order of $\Theta$.*

---

[1]See Appendix G for a discussion of how our algorithms and analyses handle such ties.

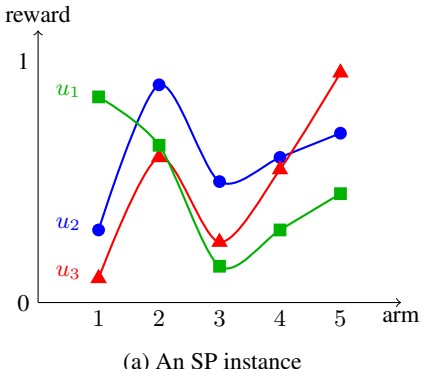
(a) An SP instance

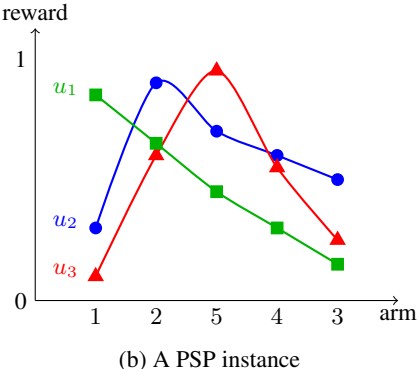
(b) A PSP instance

Figure 1: Illustration of PSP and SP instances. Each subfigure consists of curves representing the expected rewards of user-arm pairs. The instance in Figure 1a is SP, since if we reorder the arms, we get the PSP instance in Figure 1b .

Note that any PSP matrix is also SP, with the identity order serving as its SP order. In contexts where the relevant order matters, we shall write that $\Theta$ is SP w.r.t. $\prec$. Furthermore, we say that a CBR instance is (P)SP if its expected reward matrix $\Theta$ is (P)SP. The SP order need not be unique; for example, the reverse order of any SP order is also an SP order.

To illustrate the PSP and SP properties, consider Figure 1. The instance comprises three users and five arms, and each subfigure illustrates the expected rewards for each user. The instance in Figure 1a is not PSP, since there are "valleys" that contradict Inequality (1). If we reorder the arms according to the order $\prec$ such that $1 \prec 2 \prec 5 \prec 4 \prec 3$, we obtain Figure 1b. Indeed, we observe a unimodal preference shape for each user, indicating that the matrix is PSP. Thus, the instance in Figure 1a is SP and $\prec$ is its SP order.

## 3 STATISTICAL HARDNESS OF SINGLE-PEAKED INSTANCES

While SP instances are highly structured, they remain statistically as challenging to learn. To demonstrate this, we establish a regret lower bound for SP instances of $\Omega(U\sqrt{TK})$, which matches the guarantee of Corollary 2 for general preferences. Moreover, we provide a comparable lower bound for a more lenient case, where both the SP order and the users' peaks are known to the learner in advance. We do so by constructing families of hard instances where the known structural properties are fixed, and only the cardinal values differ. Since the structural information is identical across instances, the learner cannot leverage it to distinguish between them. We present these lower bounds in Theorem 3.

**Theorem 3.** *For any algorithm, the worst-case regret over SP instances is $\Omega(U\sqrt{TK})$, and $\Omega(\max\{U\sqrt{T}, \sqrt{TK}\})$ when the SP order and user peaks are known.*

Theorem 3 indicates that the SP assumption does not simplify the statistical aspect of the learning problem. However, as we demonstrate in the subsequent sections, it does affect the computational aspect–it enables efficient learning algorithms that achieve sublinear regret, standing in sharp contrast to the intractability of the general case discussed in Section 2.2.

## 4 OFFLINE ALGORITHM FOR SP INSTANCES

In this section, we present an efficient and optimal algorithm for computing $\arg\max_{\pi \in \Pi} V(\pi; \Theta)$ for SP instances. W.l.o.g., we shall assume that the SP order is the identity order, i.e., the instance is PSP. Indeed, an SP order can be extracted using our EXTRACT-ORDER algorithm (see Algorithm 2) with $\varepsilon = 0$ in $O(UK^2)$ time, after which the columns can be reordered accordingly.

The main observation used by our algorithm relates to the optimal matching conditioned on a given subset of arms. Namely, we show that if the set of selected arms is $S$, the optimal matching assigns each user to the arm in $S$ that is closest to their peak. We formalize this in Lemma 4.

**Lemma 4.** *Fix any PSP matrix $\Theta$ with peaks $p(\cdot)$, and any arm subset $S = \{k_1, \ldots, k_m\} \subseteq K$ with $k_1 < \cdots < k_m$. Let $\pi^\star \in \arg\max_{\pi, \, \mathrm{Im}(\pi) \subseteq S} V(\pi; \Theta)$. For any user $u$, if $k_j \leq p(u) \leq k_{j+1}$ for some $j$ with $1 \leq j < m$, then $\pi^\star(u) \in \{k_j, k_{j+1}\}$; otherwise, $\pi^\star(u) \in \{k_1, k_m\}$.*

Lemma 4 guides the design of a dynamic programming-based approach for finding the optimal matching for a given PSP matrix $P$, costs $c(\cdot)$, and budget $B$. We call this algorithm SP-MATCHING and explain its essence here, deferring full details to Appendix D. The algorithm maintains a table $F(k, b)$ representing the maximum achievable reward when arm $k$ is the rightmost *selected* arm, the total budget is at most $b$, and we consider only users $u$ such that $p(u) \leq k$. We add an auxiliary degenerate arm 0 such that $c_0 = 0$ and $P_{u,0} = 0$ for every user $u$, and initialize $F(0, b) = 0$ for every $b \in \{0\} \cup [B]$.

The recurrence relation for $F(k, b)$ follows from Lemma 4. When $k$ is the arm with the highest index among the selected arms and $i$ is the second highest, users with peaks before $i$ are unaffected by the inclusion of arm $k$, while those with peaks between $i$ and $k$ are assigned to whichever of the two arms offers a higher reward. Consequently, $F(k, b)$ is given by

$$\max_{\substack{i:0 \leq i < k, \\ b \geq c_i + c_k}} \left[ F(i, b - c_k) + \sum_{u:i < p(u) \leq k} \max\{P_{u,i}, P_{u,k}\} \right].$$

This maximum is always well-defined since arm 0 always satisfies its constraint. The optimal value is then $V^\star = \max_{k \in [K]} \left[ F(k, B) + \sum_{u:p(u) > k} P_{u,k} \right]$, where users with peaks beyond all selected arms are assigned to the last selected arm $k$. To enable efficient computation, we precompute for all pairs $(i, k)$ the sums $\sum_{u:i < p(u) \leq k} \max\{P_{u,i}, P_{u,k}\}$ in $O(K^2 U)$ time. This allows each entry in $F$ to be filled in $O(K)$ time, and since $F$ has $KB$ entries, the total runtime is $O(K^2 U + K^2 B)$.

**Theorem 5.** *For any PSP matrix, SP-MATCHING finds an optimal matching in time $O(K^2(U + B))$.*

## 5 ONLINE ALGORITHM FOR THE KNOWN SP STRUCTURE REGIME

We now turn to the more challenging online setup. Although the underlying expected reward matrix is SP, our reward estimates are generally not guaranteed to preserve this structure. For motivation, suppose the true reward matrix is $\Theta$, but we only have access to a noisy estimate $\bar{\Theta}$. If $\bar{\Theta}$ is itself SP, we could apply known methods to extract its SP order (Bartholdi III & Trick, 1986; Escoffier et al., 2008) and use the SP-MATCHING algorithm to obtain an approximately optimal matching w.r.t. $\Theta$. However, since $\bar{\Theta}$ only approximates $\Theta$, $\bar{\Theta}$ may not be SP. Moreover, the SP order of $\Theta$, which is crucial for our offline algorithm, remains unknown and may not be recoverable directly from $\bar{\Theta}$. Therefore, leveraging the fact that the instance is SP is non-trivial.

In this section, we take a first step towards resolving this challenge by leveraging additional structural information of the problem. Specifically, we assume that both the SP order and user peaks are known in advance to the learner. Recall from Section 3 that despite this knowledge, the learning problem remains statistically non-trivial. These structural assumptions, however, enable an efficient, optimistic UCB-style algorithm that achieves $\tilde{O}(U\sqrt{TK})$ regret. Our approach involves the construction of optimistic reward estimates using *confidence sets* (Slivkins, 2019, Chapter 8.4), as detailed in Section 5.1, while preserving the single-peaked structure. Then, we use those estimates to design the algorithm described in Section 5.2.

### 5.1 CONFIDENCE SETS AND MAXIMAL MATRIX

Given the known SP order $\prec$ and user peaks $p(\cdot)$, we construct confidence sets for the expected reward matrix that contain the true parameters with high probability. Intuitively, these sets capture all matrices that are statistically plausible given the data observed so far, and naturally shrink as more data is collected. We begin by establishing notation for standard confidence bounds. Given a history $H_t$ at round $t$, we denote by $\bar{\theta}_{u,k}(t)$ and $n_{u,k}(t)$ the empirical mean reward and the number of

---

**Algorithm 1** Match-via-Maximal (MVM)

---

**Require:** Order $\prec$, peaks $p^\star(\cdot)$
1: **for** $t = 1$ **to** $T$ **do**
2:     Construct the maximal matrix $P^t$
3:     Select $\pi_t = $ SP-MATCHING$(P^t)$
4:     Observe rewards and update history $H_t$

---

pulls of the pair $(u, k)$ observed up to round $t$, respectively. The upper confidence bound is defined as $\text{UCB}_{u,k}(t) = \bar{\theta}_{u,k}(t) + \sqrt{2 \ln T / n_{u,k}(t)}$, with the lower confidence bound $\text{LCB}_{u,k}(t)$ defined analogously using subtraction. We formally define the confidence set as follows.

**Definition 3** (Confidence set). *The* confidence set $\mathcal{C}^t = \mathcal{C}^t(\prec, p, H_t)$ *contains a matrix $P$ if and only if it is consistent with $\prec$ and $p(\cdot)$, and for any $u, k$ it holds that* $\text{LCB}_{u,k}(t) \leq P_{u,k} \leq \text{UCB}_{u,k}(t)$.

The standard UCB approach requires picking the optimistic matching at every time $t$, namely

$$\pi_t \in \arg\max_{\pi \in \Pi} \max_{P \in \mathcal{C}^t} V(\pi; P). \tag{2}$$

This discrete optimization problem over a convex set may be tricky, as different matrices yield different optimal matchings. However, a structural property of $\mathcal{C}^t$ simplifies this optimization problem. As we demonstrate shortly in Lemma 6, $\mathcal{C}^t$ includes a unique *maximal* matrix $P^t$ in an element-wise sense; thus, to find the solution of Equation (2), we should only find the optimal matching w.r.t. $P^t$.

**Lemma 6.** *For any non-empty confidence set $\mathcal{C}^t(\prec, p, H_t)$, there exists a unique element-wise maximal matrix $P^t \in \mathcal{C}^t$ such that $P^t_{u,k} \geq P_{u,k}$ for all $P \in \mathcal{C}^t$ and all $u \in [U], k \in [K]$. Furthermore,*

*this matrix is given by* $P^t_{u,k} = \begin{cases} \min_{i:k \preceq i \preceq p(u)} \text{UCB}_{u,i}(t), & k \preceq p(u) \\ \min_{i:p(u) \preceq i \preceq k} \text{UCB}_{u,i}(t), & p(u) \preceq k \end{cases}$.

We refer to the above matrix as the *maximal matrix* of $\mathcal{C}^t$, and note that its structure relies on knowledge of both the order $\prec$ and the peaks $p(\cdot)$.

## 5.2    THE MATCH-VIA-MAXIMAL ALGORITHM

Next, we present a combinatorial UCB-based algorithm, which we refer to as MVM (Algorithm 1). MVM acts optimistically–in each round, it selects the optimal matching w.r.t. the corresponding maximal matrix. The regret analysis of MVM involves standard UCB-like arguments. By the construction of the confidence sets and applying Hoeffding's inequality with a union bound over all $(u, k)$ pairs and time steps, the actual expected reward matrix $\Theta$ is in $\mathcal{C}^t$ for all $t \in [T]$ with probability at least $1 - \frac{2KU}{T^3}$. Employing the clean event technique, we ignore the complementary (bad) event, which occurs with low probability. In every round $t$, we have

$$\max_{\pi \in \Pi} V(\pi; \Theta) - V(\pi_t; \Theta) \leq V(\pi_t; P^t) - V(\pi_t; \Theta) \leq \sum_{u \in [U]} 2\sqrt{2 \ln T / n_{u, \pi_t(u)}}. \tag{3}$$

Summing Inequality (3) over all rounds, we obtain the following theorem:

**Theorem 7.** *For any SP instance with known SP order $\prec$ and peaks $p(\cdot)$,* MVM *achieves regret of at most $O(U\sqrt{TK \ln T})$ with per-round runtime of $O(K^2 U + K^2 B)$.*

**Extension to partial structural knowledge**    The MVM algorithm naturally extends to settings where the SP order and user peaks are not known exactly, but are known to belong to a polynomial-sized set $\mathcal{S}$ of candidate structures. For each candidate $(\prec, p) \in \mathcal{S}$, one can construct the corresponding maximal matrix $P^t_{(\prec, p)}$ via Lemma 6 and compute its optimal matching $\pi^t_{(\prec, p)}$ using SP-MATCHING. The algorithm then selects the matching achieving the highest optimistic value: $\pi_t \in \arg\max_{(\prec, p) \in \mathcal{S}} V(\pi^t_{(\prec, p)}; P^t_{(\prec, p)})$. Since the true structure belongs to $\mathcal{S}$, the selected matching is at least as optimistic as the one corresponding to the true parameters, and the same regret analysis yields $\tilde{O}(U\sqrt{TK})$ regret. The per-round runtime becomes $O(|\mathcal{S}| \cdot (K^2 U + K^2 B))$, which remains polynomial when $|\mathcal{S}| = \text{poly}(U, K, B)$.

---

**Algorithm 2** Extract-Order

**Require:** $\tilde{P} \in \mathbb{R}^{U \times K}$, $\varepsilon > 0$
**Ensure:** An order $\prec$ s.t. $\tilde{P}$ is $2K\varepsilon$-ASP, if such exists.
1: Initialize PQ tree $T$ on $[K]$
2: **for** $u \in [U]$ **do**
3:     Let $k_1^u, \ldots, k_K^u$ be s.t $\tilde{P}_{u,k_1^u} \geq \cdots \geq \tilde{P}_{u,k_K^u}$
4:     **for** $i \in [K-1]$ s.t. $\tilde{P}_{u,k_i^u} - \tilde{P}_{u,k_{i+1}^u} > 2\varepsilon$ **do**
5:         Add the constraint $\{k_1^u, \ldots, k_i^u\}$ to $T$
6: **if** There exists some feasible order in $T$ **then**
7:     **return** some feasible order $\prec$
8: **else, return fail**

---

**Algorithm 3** EMC

**Require:** Exploration rounds $N$
**Ensure:** A sequence of feasible matchings $(\pi_t)_{t=1}^T$
1: Define $\varepsilon_N \leftarrow \sqrt{2 \ln T / N}$
2: Pull each arm $k \in [K]$ for $N$ rounds, let $\bar{\Theta}$ be the empirical mean matrix
3: $\prec \leftarrow$ EXTRACT-ORDER($\bar{\Theta}, \varepsilon_N$)
4: Construct an SP matrix $\tilde{\Theta}$ from $(\bar{\Theta}, \prec)$ via Lemma 9 with $\delta = 2K\varepsilon_N$
5: $\tilde{\pi} \leftarrow$ SP-MATCHING($\tilde{\Theta}$)
6: Play $\tilde{\pi}$ for the remaining rounds

---

# 6 ONLINE ALGORITHM FOR THE UNKNOWN SP STRUCTURE REGIME

We now turn to the more general setting, where the learner knows only that $\Theta$ is SP, but lacks knowledge of the specific underlying order or user peaks. When the order is unknown, the confidence set must include all matrices consistent with *any* valid SP order, and the optimistic approach from Section 5 can no longer be applied efficiently; we discuss this barrier formally in Section 6.4.

To tackle this challenge, we take a three-step approach. First, we define the concept of approximate single-peaked matrices that relaxes the strict SP condition while remaining amenable to analysis. Second, we present an efficient procedure to extract a plausible SP order from empirical data. Finally, we combine these tools in an explore-then-commit algorithm that achieves sublinear regret. A key insight underlying our approach is that due to estimation noise, we cannot hope to recover the exact SP order of the true matrix $\Theta$; instead, we aim to find *some* order under which the empirical estimates are approximately single-peaked, which suffices for near-optimal matching.

## 6.1 APPROXIMATE SINGLE-PEAKED MATRICES

We begin our analysis by defining the concept of approximately single-peaked (ASP) matrices.

**Definition 4.** *We say a matrix $P$ is $\delta$-approximately single-peaked (or $\delta$-ASP) w.r.t. an order $\prec$ if for every $i, j, l \in [K]$ such that $i \prec j \prec l$ and user $u \in U$, it holds that $P_{u,j} \geq \min\{P_{u,i}, P_{u,l}\} - \delta$.*

If $P$ is $\delta$-ASP w.r.t. $\prec$, we say $\prec$ is its $\delta$-*ASP order*, analogously to SP order. The parameter $\delta$ quantifies the tolerable violation of the single-peaked property. When $\delta = 0$, this condition is equivalent to the matrix being SP w.r.t. $\prec$, as we prove in the following proposition.

**Proposition 8.** *A matrix is SP w.r.t. an order $\prec$ if and only if it is $0$-ASP w.r.t. $\prec$.*

The $\delta$-ASP condition proves useful beyond the strict $\delta = 0$ case. In fact, it follows directly from Definition 4 that if $P$ is SP w.r.t. $\prec$ and $\tilde{P}$ satisfies $\|P - \tilde{P}\|_\infty \leq \delta$, then $\tilde{P}$ is $2\delta$-ASP w.r.t. $\prec$. This implies that a noisy estimate of an SP matrix remains ASP. Conversely, given any $\delta$-ASP matrix and its ASP order, we can construct a matrix that is (exact) SP with respect to the same order, with each entry differing from the original by at most $\delta$.

**Lemma 9.** *Let $\tilde{P}$ be a $\delta$-ASP matrix w.r.t. $\prec$. There exists a matrix $P$ which is SP w.r.t. $\prec$, and satisfies $\|P - \tilde{P}\|_\infty \leq \delta$.*

We prove Lemma 9 by construction, which can be implemented in $O(UK)$ time. Lemma 9 enables us to work with SP matrices even when the estimate matrix is not SP, while not sacrificing the approximation quality. However, Lemma 9 requires the ASP order $\prec$, which we do not have.

## 6.2 EXTRACT-ORDER PROCEDURE

Relating to the previous subsection, even if we know that the estimate matrix $\bar{\Theta}$ is ASP w.r.t. the SP order of $\Theta$, this order is still unknown. In what follows, we develop the EXTRACT-ORDER algorithm, which extracts *an* ASP order from $\bar{\Theta}$ (not necessarily the SP order of $\Theta$).

We start with some intuition. Finding an order boils down to solving a set of contiguity constraints—subsets of arms that must be contiguous in any valid order. To illustrate, fix an SP matrix $P$ and suppose there exists a user $u$ and a partition of $[K]$ into two sets $A_1$ and $A_2$. Let $\beta = \min_{k \in A_1} P_{u,k}$ and $\gamma = \max_{k \in A_2} P_{u,k}$. If $\beta - \gamma > 0$, then every arm in $A_1$ is preferred by user $u$ over every arm in $A_2$. In this case, any order $\prec$ that places an arm $j \in A_2$ between two arms $i, l \in A_1$ violates the 0-ASP condition (Definition 4). Hence, according to Proposition 8, $\prec$ cannot be an SP order of $P$. Thus, any SP order of $P$ places the arms in $A_1$ in a contiguous block. As a result, any order that satisfies all such contiguity constraints is an SP order of $P$.

Next, assume we have a matrix $\tilde{P}$ such that $\|P - \tilde{P}\|_\infty \leq \varepsilon$ for the SP matrix $P$. Since every entry in $\tilde{P}$ only approximates $P$, we cannot hope to discover all the contiguity constraints of $P$. For instance, if $\min_{k \in A_1} \tilde{P}_{u,k} = \beta - \varepsilon$ and $\max_{k \in A_2} \tilde{P}_{u,k} = \gamma + \varepsilon$ for the user $u$ and partition $A_1, A_2$, then $\min_{k \in A_1} \tilde{P}_{u,k} - \max_{k \in A_2} \tilde{P}_{u,k} = \beta - \varepsilon - \gamma - \varepsilon$. In the case that this expression is negative, we would not discover the contiguity constraint of $A_1$. Similarly, we might enforce incorrect constraints. Thus, we should aim to discover as many contiguity constraints as possible from $\tilde{P}$, while remaining consistent with the contiguity constraints of $P$. To that end, if a triplet $u, A_1$ and $A_2$ satisfies

$$\min_{k \in A_1} \tilde{P}_{u,k} - \max_{k \in A_2} \tilde{P}_{u,k} > 2\varepsilon, \tag{4}$$

we can safely add a contiguity constraint on $A_1$ since the proximity between $\tilde{P}$ and $P$ guarantees $\min_{k \in A_1} P_{u,k} - \max_{k \in A_2} P_{u,k} > 0$; hence, this constraint also applies to the matrix $P$.

The EXTRACT-ORDER algorithm, described in Algorithm 2, formalizes this approach using a PQ tree (Booth & Lueker, 1976), an efficient data structure that specializes in encoding and resolving contiguity constraints. PQ trees maintain sets of items that must appear consecutively in any valid order, and can be used both to determine whether a consistent order exists and to efficiently construct such an order. EXTRACT-ORDER initializes a PQ tree (Line 1), and for each user $u$, it sorts the arms by preference values (Lines 2-3). Whenever there is a significant gap (exceeding $2\varepsilon$) between consecutive preference values, which exactly corresponds to the case described in Inequality (4), it adds a contiguity constraint requiring all higher-valued arms to appear consecutively (Line 5). Finally, the algorithm returns an order on the columns of $\tilde{P}$ that satisfies all the imposed contiguity constraints, if such an order exists (Line 7). Otherwise, it fails and returns nothing (Line 8)

Notably, the constraints enforced by this procedure are not only consistent with the SP order of $P$, but also ensure bounded valley depth (in the sense of Definition 4). Formally,

**Lemma 10.** *Let $\tilde{P}$ be a matrix such that $\|\tilde{P} - P\|_\infty \leq \varepsilon$ for some SP matrix $P$.* EXTRACT-ORDER*$(\tilde{P}, \varepsilon)$ returns an order $\prec$ such that $\tilde{P}$ is $(2K\varepsilon)$-ASP w.r.t. $\prec$.*

The runtime complexity of EXTRACT-ORDER is dominated by sorting user preferences and PQ tree operations. Sorting preferences for all users requires $O(UK \log K)$ time. Subsequently, for each user, we may add up to $K-1$ contiguity constraints to the PQ tree, where each addition takes $O(K)$ time, yielding $O(UK^2)$ time for all constraints. The final feasibility check and order extraction from the PQ tree requires $O(K)$ time. Thus, the overall runtime is $O(UK^2)$.

## 6.3 EXPLORE-THEN-COMMIT ALGORITHM

We now present the Explore-then-Match-and-Commit algorithm (EMC), described in Algorithm 3, which combines the tools developed in the previous subsections to achieve sublinear regret. It operates in five phases. First, given an input parameter $N$, the algorithm explores by pulling each arm $k \in [K]$ for $N$ rounds and collects empirical means $\bar{\Theta}$. Second, it applies EXTRACT-ORDER and extracts an order $\prec$ from $\bar{\Theta}$. The parameter $\varepsilon$ is chosen so that $\|\Theta - \bar{\Theta}\|_\infty \leq \varepsilon$ holds with high probability. In the third phase, it uses $\bar{\Theta}$ and the order $\prec$ to construct the matrix $\tilde{\Theta}$, which is SP w.r.t. $\prec$ and is element-wise close to $\bar{\Theta}$. Fourth, it executes SP-MATCHING on $\tilde{\Theta}$ to find an optimal matching $\tilde{\pi}$. And in the fifth and last phase, it exploits–it plays $\tilde{\pi}$ for the remaining $T - NK$ rounds. Importantly, the computational effort in Lines 3–5 is only $O(K^2 U + K^2 B)$, ensuring that the algorithm is computationally efficient. This is in sharp contrast to the exponential runtime we obtain for general instances in Section 2.2.

The regret analysis for the commitment phase hinges on the approximation quality between the matrix $\tilde{\Theta}$ used for the matching and the true expected reward matrix $\Theta$. Since $\tilde{\Theta}$ is constructed to be close to the empirical estimates $\bar{\Theta}$, and $\bar{\Theta}$ concentrates around $\Theta$ with sufficient exploration, the value gap $V(\tilde{\pi}; \tilde{\Theta}) - V(\tilde{\pi}; \Theta)$ is bounded by the estimation errors and the approximation errors from Lemma 9 and EXTRACT-ORDER. By optimizing the choice of exploration rounds $N$, we balance the exploration cost against the quality of the resulting approximation.

**Theorem 11.** EMC$\left(N = \left\lceil T^{2/3}(\ln T)^{1/3} \right\rceil\right)$ *yields an expected regret of at most* $\tilde{O}(UKT^{2/3})$.

### 6.4 HARDNESS OF OPTIMISTIC MATCHING WITH UNKNOWN ORDER

In the known SP structure regime (Section 5), the MvM algorithm achieves $\tilde{O}(U\sqrt{TK})$ regret by solving the optimistic matching problem in each round. A natural question is whether a similar approach can yield improved regret guarantees when the SP structure is unknown. Recall that the optimistic approach requires solving Equation (2), where $\mathcal{C}^t$ is now the confidence set of statistically plausible SP matrices. When the order and peaks are known, Lemma 6 shows that $\mathcal{C}^t$ admits a unique element-wise maximal matrix, reducing the optimization to a single call to SP-MATCHING. However, when the order is unknown, $\mathcal{C}^t$ must include all SP matrices consistent with *any* valid order, and no such maximal element exists in general.

We note that if the optimistic matching could be solved efficiently in this regime, the same analysis as in Theorem 7 would yield the same $\tilde{O}(U\sqrt{TK})$ regret guarantee. One might hope that a more sophisticated algorithm could achieve this. We show that this is unlikely by proving that even a simpler subproblem–finding the best matrix in $\mathcal{C}^t$ for a *fixed* matching–is NP-hard to approximate.

**Theorem 12.** *Consider the following optimization problem, termed* MAX-SP-WCS*: given sets of users $U$ and arms $K$, a matching $\pi : U \to K$, and confidence intervals $[\text{LCB}_{u,k}, \text{UCB}_{u,k}]$ for each $(u,k) \in U \times K$, find $\max_{P \in \mathcal{C}} V(\pi; P)$, where $\mathcal{C}$ contains all SP matrices respecting the confidence intervals. Then, approximating* MAX-SP-WCS *within a factor of $\frac{3}{4} + \delta$ is NP-hard for any $\delta > 0$.*

## 7 DISCUSSION

We studied a budgeted matching problem under single-peaked preferences. While the general offline problem is NP-hard to approximate within a constant factor, imposing a single-peaked structure enables polynomial-time optimization. Nevertheless, we demonstrated that this structure does not trivialize the statistical aspect of the learning problem, which remains as challenging as the general preferences case. When the SP structure is unknown, we introduce an explore-then-commit approach that first extracts an approximate order, then commits to a near-optimal policy, yielding $\tilde{O}(UKT^{2/3})$ regret. For the more lenient case of known SP order and user peaks, we developed an optimistic algorithm based on maximal matrices that achieves $\tilde{O}(U\sqrt{TK})$ regret. We also conducted an experimental validation, which is deferred to Appendix H due to space constraints.

We identify several interesting directions for future work. First, comparing our algorithmic results with the lower bounds, we observe room for improvement. For the unknown SP structure regime, EMC achieves $\tilde{O}(UKT^{2/3})$ regret, while the $\Omega(U\sqrt{TK})$ lower bound is statistically tight (achievable by inefficient algorithms). It remains open whether the $\sqrt{T}$ rate can be achieved efficiently, or if our $T^{2/3}$ rate is tight for polynomial-time algorithms. Theorem 12 provides evidence toward the latter: exponentially many SP structures may remain consistent with observations, and we show that even evaluating the optimistic value for a fixed matching is NP-hard to approximate. For known order and peaks, MvM achieves $\tilde{O}(U\sqrt{TK})$ regret (Theorem 7), exceeding the lower bound by $\sqrt{\ln T} \cdot \max\{\sqrt{K}, U\}$. Future work could resolve this gap.

Second, our results suggest that single-peaked preferences could potentially simplify other computationally challenging problems in online learning, suggesting an alternative route to a solution rather than $\alpha$-regret. Finally, extending our results to more complex preference structures (Sliwinski & Elkind, 2019; Peters & Lackner, 2020) presents an intriguing challenge.

### ACKNOWLEDGMENTS

This research was supported by the Israel Science Foundation (ISF; Grant No. 3079/24).

REPRODUCIBILITY STATEMENT

All theoretical results are supported by complete proofs provided in the appendices. Algorithms are fully specified in the main text and appendices, including pseudocode and complexity analyses. The assumptions required for our results are explicitly stated and discussed in the problem definition and algorithmic sections. Our experimental evaluation is based on synthetic single-peaked preference matrices generated according to a well-defined random procedure, with details given in Appendix H. The source code implementing all algorithms and experiments, along with instructions for reproducing the figures, is attached to the submission and will be released upon publication. The experiments rely only on standard Python libraries and require modest computational resources (a single CPU).

ETHICS STATEMENT

Our work is theoretical, and the experimental evaluation relies only on synthetic data, involving no human subjects or sensitive information. The results may inform the design of recommendation and matching systems under budget constraints. Applying such methods in sensitive domains (e.g., political content or hiring) requires careful consideration of fairness and user autonomy.

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

---

**Algorithm 4** Sep-MvM

---

1: Initialize $n_{u,k} \leftarrow 0, \bar{\theta}_{u,k} \leftarrow 0$ for all $u, k$.
2: *// Phase 1: Exploration with Monitoring*
3: **for** $t = 1, \dots, \tau = KT^{2/3}$ **do**
4:     Select arm $k_t = (t \pmod{K}) + 1$ for all users (Round-Robin).
5:     Observe rewards, update empirical means $\bar{\theta}_{u,k}$ and counts $n_{u,k}$.
6:     Update Confidence Intervals: $\mathcal{I}_{u,k}(t) = [\bar{\theta}_{u,k} - \sqrt{\frac{2 \ln T}{n_{u,k}}}, \bar{\theta}_{u,k} + \sqrt{\frac{2 \ln T}{n_{u,k}}}]$.
7:     SEP $\leftarrow \forall u, \forall i \neq j, \mathcal{I}_{u,i}(t) \cap \mathcal{I}_{u,j}(t) = \emptyset$.
8:     **if** SEP is True **then**
9:         Break Phase 1., proceed to Phase 2.
10: *// Phase 2: Structure Exploitation*
11: **if** SEP is True (Structure Recovered) **then**
12:     Recover SP Order $\prec$ using EXTRACT-ORDER($\bar{\Theta}, 0$) and the peaks.
13:     Run MVM using recovered order $\prec$ and inferred peaks.
14: **else**
15:     *// Timeout Reached - fallback to EMC Strategy*
16:     Solve $\bar{\pi} \leftarrow$ SP-MATCHING($\hat{\Theta}$) (using the commitment phase of EMC).
17:     Commit to $\bar{\pi}$ for rounds $t, \dots, T$.

---

# A EXTENSIONS

In this appendix, we present two extensions of our main results. First, we extend the MVM algorithm to handle instances with unknown structure but statistically simpler learning due to distinct preference values (Section A.1). Second, we generalize the EMC algorithm to accommodate non-single-peaked instances (Section A.2).

## A.1 SEPARATED INSTANCES

We improve our guarantees for instances that are statistically *simpler* to learn due to distinct preference values. Specifically, we consider *separated instances*, where we quantify separation by $\Delta_{\text{global}} = \min_u \min_{i \neq j} |\theta_{u,i} - \theta_{u,j}|$.

**Algorithm overview.** The SEP-MVM algorithm, presented in Algorithm 4, follows a hybrid exploration-exploitation strategy. It begins by exploring all arms in a round-robin manner, maintaining confidence intervals $\mathcal{I}_{u,k}(t) = [\text{LCB}_{u,k}(t), \text{UCB}_{u,k}(t)]$ for each user-arm pair. The algorithm seeks a *separation event* before a timeout $\tau = KT^{2/3}$. Separation occurs when, for each user, the confidence intervals of all arms are pairwise disjoint. A direct concentration analysis reveals that such separation must occur after at most $O(K \ln T / \Delta_{\text{global}}^2)$ rounds. After separation, the algorithm reconstructs the SP order using EXTRACT-ORDER and switches to MVM with the estimated order and peaks. If separation is not achieved by $\tau$, the algorithm infers that the instance is not well-separated and reverts to EMC.

**Proposition 13.** SEP-MVM *yields a regret of at most* $\tilde{O}\left(U \min\left\{KT^{2/3}, \sqrt{TK} + K/\Delta_{\text{global}}^2\right\}\right)$, *for* $\Delta_{\text{global}} = \min_u \min_{i \neq j} |\theta_{u,i} - \theta_{u,j}|$, *with a per-step computational complexity of* $O(K^2 U + K^2 B)$.

*Proof.* We analyze the regret conditioned on the standard clean event $\mathcal{E} = \{\forall u, k, t : |\hat{\theta}_{u,k}(t) - \theta_{u,k}| \leq \sqrt{\frac{2 \ln T}{n_{u,k}(t)}}\}$, which holds with probability at least $1 - 2UK/T^3$. Under $\mathcal{E}$, the true mean is always contained within the confidence intervals $\mathcal{I}_{u,k}(t)$.

We consider two cases based on the magnitude of the instance gap $\Delta_{\text{global}}$.

**Case 1:** $\Delta_{\text{global}} < \sqrt{\frac{32 \ln T}{T^{2/3}}}$. First, notice that if separation does occur before the timeout $\tau = KT^{2/3}$, then the regret will be at most $O(UKT^{2/3})$, as it comprises of the exploration regret (which will be less that $KT^{2/3}$ rounds) and the regret of the MVM algorithm, which is is at

most $O(U\sqrt{TK})$ when run with the correct parameters. Next, suppose the separation condition is *not* met by the timeout. The algorithm switches to the EMC commitment strategy, after performing its exact exploration phase with the same exploration parameter as in Theorem 11. Thus, the same regret bound applies, and we get $R(T) \leq \tilde{O}(UKT^{2/3})$. And indeed, when the condition $\Delta_{\text{global}} < \sqrt{\frac{32 \ln T}{T^{2/3}}}$ holds, this is the smallest of the two expressions in the minimum (up to poly-logarithmic factors).

**Case 2:** $\Delta_{\text{global}} \geq \sqrt{\frac{32 \ln T}{T^{2/3}}}$. In this case, separation occurs *before* the timeout. To see this, notice that under the clean event, separation must occur if $\sqrt{\frac{2 \ln T}{n_{u,k}}} \leq \Delta_{\text{global}}/4$. Since samples are collected round-robin, $n_{u,k}(t) \approx t/K$. The condition requires:

$$\sqrt{\frac{2K \ln T}{t}} < \frac{\Delta_{\text{global}}}{4} \implies t > \frac{32K \ln T}{\Delta_{\text{global}}^2}.$$

Thus, the separation event occurs at $t_{sep} = O(\frac{K \ln T}{\Delta_{\text{global}}^2}) < \tau$.

After the separation, the structure can be recovered perfectly (again, conditioning on the clean event)—pairwise disjointness implies the sorted order of empirical means is identical to the sorted order of true means. Thus, EXTRACT-ORDER will recover a true SP order and peaks.

After $t_{sep}$, we switch to MVM. The total regret is the sum of the exploration regret (up to $t_{sep}$) and the MVM regret (from $t_{sep}$ to $T$):

$$R(T) \leq \underbrace{U \cdot t_{sep}}_{\text{Exploration}} + \underbrace{R_{\text{MVM}}(T - t_{sep})}_{\text{MVM Regret}}$$

Substituting $t_{sep} = \tilde{O}(K/\Delta_{\text{global}}^2)$ and the MVM bound $\tilde{O}(U\sqrt{TK})$:

$$R(T) \leq \tilde{O}\left(\frac{UK}{\Delta_{\text{global}}^2} + U\sqrt{TK}\right).$$

**Conclusion.** Taking the minimum of the two cases (since the algorithm automatically executes the better strategy via the timeout mechanism), we obtain the bound:

$$\tilde{O}\left(U \min\left\{KT^{2/3}, \sqrt{TK} + \frac{K}{\Delta_{\text{global}}^2}\right\}\right).$$

$\square$

## A.2 BEYOND SINGLE-PEAKED INSTANCES

We extend EMC to work on non-single-peaked instances. The key idea is that even when the true preference matrix $\Theta$ is not SP, we can still apply our algorithmic framework and obtain meaningful guarantees that degrade gracefully with the distance from the nearest SP instance.

**Algorithm overview.** The NSP-EMC algorithm, presented in Algorithm 5, maintains the same explore-then-commit structure as EMC but adds a binary search procedure to find the minimal tolerance parameter $\varepsilon$ for which EXTRACT-ORDER succeeds. This adaptive approach allows the algorithm to handle instances that are approximately single-peaked without requiring prior knowledge of the approximation quality.

**Proposition 14.** *Fix an instance $\mathcal{I}$ (possibly not SP), and let $\Theta$ be its expectation matrix. Then, NSP-EMC achieves a regret of at most $O\left(UKT^{2/3}(\ln T)^{1/3} + \gamma UKT\right)$, where $\gamma = \inf_{P \text{ is SP}} \|\Theta - P\|_\infty$, with a computational complexity of $O(UK^2 \log(T) + K^2 B)$.*

*Proof.* Let $P^* \in \mathcal{S}$ be an SP matrix such that $\|\Theta - P^*\|_\infty = \gamma$, and let $\Delta_{est} = \sqrt{2 \ln T/N}$. We condition on the clean event $\mathcal{E} = \{\|\overline{\Theta} - \Theta\|_\infty \leq \Delta_{est}\}$, which holds with probability at least $1 - 2UK/T^4$ by Hoeffding's inequality and a union bound.

---

**Algorithm 5** NSP-EMC

---

**Require:** Exploration rounds $N$
 1: Match each user $u$ to each arm $k \in [K]$ for $N$ rounds; let $\overline{\Theta}$ be the empirical mean matrix.
 2: **Binary Search for $\varepsilon$:**
 3: Set $\varepsilon_{low} \leftarrow 0, \varepsilon_{high} \leftarrow 1, \hat{\varepsilon} \leftarrow 1, \prec \leftarrow$ Identity
 4: **while** $\varepsilon_{high} - \varepsilon_{low} > \frac{1}{T}$ **do**
 5: $\quad \varepsilon_{mid} \leftarrow (\varepsilon_{low} + \varepsilon_{high})/2$
 6: $\quad \prec_{temp} \leftarrow$ EXTRACT-ORDER$(\overline{\Theta}, \varepsilon_{mid})$
 7: $\quad$ **if** $\prec_{temp} \neq$ fail **then**
 8: $\quad\quad \hat{\varepsilon} \leftarrow \varepsilon_{mid}$
 9: $\quad\quad \prec \leftarrow \prec_{temp}$
10: $\quad\quad \varepsilon_{high} \leftarrow \varepsilon_{mid}$
11: $\quad$ **else**
12: $\quad\quad \varepsilon_{low} \leftarrow \varepsilon_{mid}$
13: Construct SP matrix $\tilde{\Theta}$ from $(\overline{\Theta}, \prec)$ via Lemma 9 with $\delta = 2K\varepsilon_{high}$.
14: $\tilde{\pi} \leftarrow$ SP-MATCHING$(\tilde{\Theta})$
15: Play $\tilde{\pi}$ for the remaining $T - NK$ rounds.

---

**Step 1: Feasibility of the Binary Search.** By the triangle inequality and the definition of $\gamma$:

$$\|\overline{\Theta} - P^*\|_\infty \leq \|\overline{\Theta} - \Theta\|_\infty + \|\Theta - P^*\|_\infty \leq \Delta_{est} + \gamma.$$

Let $\varepsilon^* = \Delta_{est} + \gamma$. Since $P^*$ is an SP matrix (and thus valid with respect to some SP order $\prec_{P^*}$), Lemma 10 guarantees that EXTRACT-ORDER$(\overline{\Theta}, \varepsilon^*)$ will succeed and return a valid order. Because the binary search finds the minimal feasible $\varepsilon$ (up to precision $1/T$) for which EXTRACT-ORDER returns a valid order (note that this condition is monotone in $\varepsilon$), the found parameter $\hat{\varepsilon}$ satisfies:

$$\varepsilon_{high} \leq \varepsilon^* + \frac{2}{T} = \Delta_{est} + \gamma + \frac{2}{T}.$$

We omit the $\frac{2}{T}$ term in the remainder of the analysis, as it is negligible in the final bound.

**Step 2: Bounding the Approximation Error.** The algorithm produces an order $\prec$ using $\varepsilon_{high}$. According to Lemma 10, the empirical matrix $\overline{\Theta}$ is $(2K\varepsilon_{high})$-ASP with respect to $\prec$. Subsequently, the algorithm constructs $\tilde{\Theta}$ using Lemma 9 with $\delta = 2K\varepsilon_{high}$. Lemma 9 guarantees that $\tilde{\Theta}$ is SP with respect to $\prec$ and satisfies:

$$\|\tilde{\Theta} - \overline{\Theta}\|_\infty \leq 2K\varepsilon_{high}.$$

Substituting the bound for $\varepsilon_{high}$ from Step 1:

$$\|\tilde{\Theta} - \overline{\Theta}\|_\infty \leq 2K(\Delta_{est} + \gamma).$$

**Step 3: Total Value Gap and Regret.** We bound the difference between the true optimal matching $\pi^*$ and our committed matching $\tilde{\pi}$. Since $\tilde{\pi}$ is optimal for $\tilde{\Theta}$:

$$V(\pi^*; \Theta) - V(\tilde{\pi}; \Theta) = V(\pi^*; \Theta) - V(\pi^*; \tilde{\Theta}) + V(\pi^*; \tilde{\Theta}) - V(\tilde{\pi}; \Theta)$$
$$\leq V(\pi^*; \Theta) - V(\pi^*; \tilde{\Theta}) + V(\tilde{\pi}; \tilde{\Theta}) - V(\tilde{\pi}; \Theta) \quad \text{(optimality of } \tilde{\pi} \text{ on } \tilde{\Theta})$$
$$\leq 2U\|\Theta - \tilde{\Theta}\|_\infty.$$

Using the triangle inequality:

$$\|\Theta - \tilde{\Theta}\|_\infty \leq \|\Theta - \overline{\Theta}\|_\infty + \|\overline{\Theta} - \tilde{\Theta}\|_\infty$$
$$\leq \Delta_{est} + 2K(\Delta_{est} + \gamma)$$
$$= (2K + 1)\Delta_{est} + 2K\gamma.$$

Thus, the per-round regret during the commitment phase is bounded by:

$$V(\pi^*; \Theta) - V(\tilde{\pi}; \Theta) \leq 2U(2K + 1)\Delta_{est} + 4UK\gamma.$$

**Step 4: Total Regret.** Substituting $N = \lceil T^{2/3}(\ln T)^{1/3} \rceil$ and $\Delta_{est} = \sqrt{2 \ln T/N}$:

$$R_T \leq NKU + T\left(2U(2K+1)\sqrt{\frac{2\ln T}{N}} + 4UK\gamma\right) \leq O(UKT^{2/3}(\ln T)^{1/3} + \gamma UKT).$$

**Computational Complexity.** The exploration phase requires $O(NK) = O(KT^{2/3}(\ln T)^{1/3})$ rounds. The binary search performs $O(\log T)$ calls to EXTRACT-ORDER, each taking $O(UK^2)$ time, for a total of $O(UK^2 \log T)$. The final SP-MATCHING call takes $O(K^2(U+B))$ time. Thus, the total computational complexity is $O(UK^2 \log T + K^2 B)$. $\qquad\square$

# B    GENERAL PREFERENCES ANALYSIS

In this appendix, we provide a comprehensive analysis of the computational complexity of achieving sublinear regret under general preferences. We first establish the fundamental hardness of the offline problem, then derive its implications for online learning, and finally discuss the $\alpha$-regret framework as a tractable alternative.

## B.1    HARDNESS OF THE OFFLINE PROBLEM

We begin by proving that the offline budgeted matching problem is NP-hard to approximate within any factor better than $(1 - 1/e)$.

**Theorem 1.** *It is NP-hard to approximate* $\arg\max\limits_{\pi\in\Pi} V(\pi; \Theta)$ *within any factor better than* $(1-1/e)$.

**Proof of Theorem 1**. We proceed via reduction from the MAX $\ell$-COVER problem. In this problem, one is given a universe $\mathcal{U}$ of elements, a collection of subsets $\mathcal{S} = \{S_1, \ldots, S_m\}$, and an integer $\ell$. The objective is to select at most $\ell$ subsets whose union covers the maximum number of elements in $\mathcal{U}$. This problem is known to be NP-hard to approximate better than $(1 - 1/e)$ (Feige, 1998).

Given an instance of the Max $\ell$-Cover problem, we construct a CBR instance with $U = |\mathcal{U}|$ users (representing elements) and $K = m$ arms (representing subsets). We set unit costs $c_k = 1$, budget $B = \ell$, and define preferences such that $\Theta_{u,k} = 1$ if element $u \in S_k$ and 0 otherwise.

In this constructed instance, each matching $\pi$ induces a set of selected arms $\mathcal{K} = \{k : \exists u \in U \text{ s.t. } \pi(u) = k\}$. A user $u$ contributes 1 to the total reward only if assigned to a selected arm $k$ where $u \in S_k$. Accordingly, given a fixed $\mathcal{K}$, the optimal matching allocates each user to a covering item if one exists; otherwise, the user yields zero reward. This implies that maximizing total reward is equivalent to maximizing coverage using at most $\ell$ items. It follows that any algorithm approximating the offline matching problem better than $(1-1/e)$ would violate the inapproximability result for Max $\ell$-Cover. $\qquad\square$

## B.2    COMPUTATIONAL BARRIER FOR EFFICIENT ONLINE LEARNING

The NP-hardness of approximating the offline problem beyond $(1 - 1/e)$ directly implies that efficient online learning with sublinear regret is impossible under standard complexity assumptions.

**Corollary 15.** *For any $U, K, B$ and $T = \text{poly}(U, K, B)$, any poly-time online algorithm achieves regret $\Omega(T)$, unless $\text{P} = \text{NP}$.*

**Proof of Corollary 15**. Assume towards contradiction that there exists an algorithm $\mathcal{A}$ running in time $\text{poly}(U, K, B)$ per round that achieves regret $R_T \leq CT^\rho$ for some constants $\rho \in [0, 1)$ and $C > 0$, for horizons $T$ that are polynomial in $U, K, B$. For ease of presentation, we assume $T = U^\alpha K^\beta B^\gamma$ for some $\alpha, \beta, \gamma \geq 1$.

**Restriction to hard instances.** By Theorem 1, the inapproximability result holds already for instances produced by the reduction from Max-$\ell$-Cover Feige (1998). These instances have $\Theta \in \{0, 1\}^{U \times K}$ with unit costs, budget $B = \ell$, and at least one entry equal to 1. Consequently, for such instances we always have $\max_{\pi\in\Pi(\Theta)} V(\pi; \Theta) \geq 1$. Since the hardness persists on this restricted subclass, it suffices to prove the corollary under the additional assumption $\max_{u,k} \theta_{u,k} = 1$.

**Construction of the online instance.** Given such an offline instance with expected reward matrix $\Theta$, we construct a deterministic online instance where the realized reward of each $(u, k)$ pair is fixed to $\theta_{u,k}$. Additionally, we pick an arbitrary arm and duplicate it so that there are

$$D = \max\left\{1, \left\lceil (10C)^{\frac{1}{\beta(1-\rho)}} U^{\frac{\alpha(\rho-1)}{\beta(1-\rho)}} B^{\frac{\gamma(\rho-1)}{\beta(1-\rho)}} \right\rceil \right\}$$

copies of it. Let $\Theta'$ denote the resulting matrix. Duplicating an arm cannot increase the optimal offline value, since multiple users can already be assigned to the same arm at no additional per-round cost. Therefore,

$$\max_{\pi \in \Pi(\Theta')} V(\pi; \Theta') = \max_{\pi \in \Pi(\Theta)} V(\pi; \Theta) = \text{OPT} \geq 1.$$

**Simulating $\mathcal{A}$.** Run $\mathcal{A}$ for $T' = U^\alpha D^\beta B^\gamma$ rounds on the expanded instance, obtaining matchings $\pi^1, \ldots, \pi^{T'}$, and return

$$\pi^{\text{best}} = \arg\max_{t \in [T']} V(\pi^t; \Theta').$$

The regret guarantee yields

$$T' \cdot \text{OPT} - \sum_{t=1}^{T'} V(\pi^t; \Theta') \leq CT'^\rho.$$

Since $V(\pi^{\text{best}}; \Theta') \geq \frac{1}{T'} \sum_{t=1}^{T'} V(\pi^t; \Theta')$, it follows that

$$\text{OPT} - V(\pi^{\text{best}}; \Theta') \leq CT'^{\rho-1}.$$

**Approximation guarantee.** By construction of $D$, we ensured that $CT'^{\rho-1} \leq 0.1$. Hence

$$\text{OPT} - V(\pi^{\text{best}}; \Theta') \leq 0.1 \leq 0.1\text{OPT}.$$

Since any matching on the duplicated instance is equivalent to a matching on the non-duplicated one, without loss of generality, we assume that $\pi^{\text{best}} \in \Pi(\Theta)$. Thus:

$$V(\pi^{\text{best}}; \Theta) \geq 0.9\, V(\pi^\star; \Theta).$$

**Contradiction.** The algorithm above runs in $\text{poly}(U, K, B)$ time and produces a 0.9-approximation to the offline optimum, contradicting the $(1 - 1/e)$-hardness result from Theorem 1 unless P = NP. $\qquad\square$

### B.3 OPTIMAL REGRET WITHOUT COMPUTATIONAL CONSTRAINTS

If we disregard computational constraints, optimal regret bounds can be attained by applying the standard CUCB algorithm (Chen et al., 2013), assuming access to an exact optimization oracle.

**Corollary 2.** *There exists a UCB-based algorithm that achieves a regret of $O(U\sqrt{KT \log T})$, which is optimal up to logarithmic factors.*

**Proof of Corollary 2.** The CUCB algorithm (Chen et al., 2013) maintains averages $\bar{\theta}_{u,k}(t)$ and counts $n_{u,k}(t)$ for each user-arm pair. In each round $t$, it computes UCB estimates $P_{u,k}(t) = \bar{\theta}_{u,k}(t) + \sqrt{2 \ln T / n_{u,k}(t)}$ (or $\infty$ if $n_{u,k}(t) = 0$) and selects $\pi_t \in \arg\max_{\pi \in \Pi} V(\pi; P(t))$ using an optimization oracle. We note that we slightly changed the exact specification of the constants of CUCB, but it is of the same spirit.

By Hoeffding's inequality and a union bound over all $(u, k)$ pairs and time steps, the clean event

$$\mathcal{E} = \{\theta_{u,k} \in [\text{LCB}_{u,k}(t), \text{UCB}_{u,k}(t)] \text{ for all } u, k, t\}$$

holds with probability at least $1 - 2UK/T^3$. We condition on $\mathcal{E}$; its complement contributes negligible regret.

Under $\mathcal{E}$, since $P(t)$ is element-wise maximal over all matrices respecting the confidence bounds, optimistic selection ensures that for any round $t$:

$$\max_{\pi \in \Pi} V(\pi; \Theta) - V(\pi_t; \Theta) \leq V(\pi_t; P(t)) - V(\pi_t; \Theta)$$

$$\leq \sum_{u \in [U]} 2\sqrt{\frac{2 \ln T}{n_{u, \pi_t(u)}(t)}}. \tag{5}$$

The first inequality uses optimism: $V(\pi_t; P(t)) \geq V(\pi^\star; P(t)) \geq V(\pi^\star; \Theta)$, where $\pi^\star$ is the optimal matching. The second inequality bounds the gap by the sum of confidence widths.

Summing Inequality 5 over all rounds and regrouping by arm pulls:

$$R_T \leq O\left( \sqrt{\ln T} \sum_{u \in [U]} \sum_{k \in [K]} \sum_{j=1}^{n_{u,k}(T)} \frac{1}{\sqrt{j}} \right)$$

$$= O\left( \sqrt{\ln T} \sum_{u \in [U]} \sum_{k \in [K]} \sqrt{n_{u,k}(T)} \right),$$

where we used the standard bound $\sum_{j=1}^{n} 1/\sqrt{j} \leq 2\sqrt{n}$. Applying Jensen's inequality to the inner sum yields $\sum_k \sqrt{n_{u,k}(T)} \leq \sqrt{K \sum_k n_{u,k}(T)} = \sqrt{KT}$. Summing over all $U$ users gives $R_T = O(U\sqrt{TK \ln T})$.

The matching lower bound $\Omega(U\sqrt{TK})$ follows from Theorem 3, as SP instances form a subclass of general instances. $\qquad\square$

While this algorithm is statistically optimal, it assumes an optimization oracle that must solve an NP-hard problem at every time step, rendering it computationally impractical for general instances.

### B.4 EFFICIENT APPROXIMATION VIA ALPHA-REGRET

To bridge the gap between computational efficiency and performance guarantees, we consider the $\alpha$-regret framework, which compares against the best efficiently computable solution rather than the true optimum.

**Definition 5** (Alpha Regret). *For an approximation factor $\alpha \in (0, 1]$, the expected cumulative $\alpha$-regret over $T$ rounds is defined as:*

$$R_\alpha(T) = \alpha \cdot T \cdot \max_{\pi \in \Pi} V(\pi; \Theta) - \mathbb{E}\left[ \sum_{t=1}^{T} \sum_{u \in [U]} r_{u, \pi^t(u)}^t \right].$$

By reformulating our problem as submodular maximization subject to knapsack constraints, we can leverage existing approximation algorithms to achieve efficient learning with meaningful guarantees.

**Lemma 16.** *The reformulated reward function*

$$f(S) = \sum_{u \in [U]} \max_{k \in S} \Theta_{u,k}$$

*is submodular and monotone.*

**Proof of Lemma 16.** We prove both properties separately.

*Submodularity.* For $M \subseteq N \subseteq [K]$ and $k \in [K] \setminus N$, we need

$$f(M \cup \{k\}) - f(M) \geq f(N \cup \{k\}) - f(N).$$

Let $\theta_u^M = \max_{j \in M} \Theta_{u,j}$ and $\theta_u^N = \max_{j \in N} \Theta_{u,j}$. Since $M \subseteq N$, we have $\theta_u^M \leq \theta_u^N$. The marginal gains are:

$$f(M \cup \{k\}) - f(M) = \sum_{u \in [U]} \max\{0, \Theta_{u,k} - \theta_u^M\},$$

Table 1: Reward structure for the base case (Case 1). Costs are shown in parentheses.

| | Arm 1 ($c_1 = 1$) | Arm 2 ($c_2 = 3$) | Arm 3 ($c_3 = 1$) | Arm 4 ($c_4 = 1$) |
|---|---|---|---|---|
| User 1 | $\mu$ | 1 | $\nu$ | 0 |
| User 2 | 0 | 0 | 0 | 1 |
| User 3 | 0 | 0 | 0 | 1 |

$$f(N \cup \{k\}) - f(N) = \sum_{u \in [U]} \max\{0, \Theta_{u,k} - \theta_u^N\}.$$

For each user $u$, since $\theta_u^M \leq \theta_u^N$:

$$\max\{0, \Theta_{u,k} - \theta_u^M\} \geq \max\{0, \Theta_{u,k} - \theta_u^N\}.$$

Summing over all users gives the desired inequality.

*Monotonicity.* For $M \subseteq N$, since $\max_{k \in M} \Theta_{u,k} \leq \max_{k \in N} \Theta_{u,k}$ for each user $u$, we have:

$$f(M) = \sum_{u \in [U]} \max_{k \in M} \Theta_{u,k} \leq \sum_{u \in [U]} \max_{k \in N} \Theta_{u,k} = f(N).$$

$\square$

**Proposition 17.** *There exists an algorithm running in $O(K^3 U)$ time per-round achieving expected cumulative $\frac{1}{2}$-regret of:*

$$\mathbb{E}[R_{1/2}(T)] \leq \tilde{O}\left(U(K + \beta)^{4/3} T^{2/3}\right),$$

*where $\beta = {}^B/_{\min_k c_k}$ is the budget-to-minimum-cost ratio.*

**Proof of Proposition 17**. *Regret analysis.* We reformulate our problem as submodular maximization over subsets, where we select $S \subseteq [K]$ subject to the budget constraint $\sum_{k \in S} c_k \leq B$ and each user receives their most preferred item from the selected subset. The reformulated reward function $f(S) = \sum_{u \in [U]} \max_{k \in S} \Theta_{u,k}$ is submodular and monotone by Lemma 16. Under knapsack constraints, the Greedy+Max algorithm of Yaroslavtsev et al. (2020) achieves a $\frac{1}{2}$-approximation and is $\left(\frac{1}{2}, \frac{1}{2} + \tilde{K} + 2\beta\right)$-robust in the sense of Nie et al. (2023), where $\tilde{K} = \min\{K, B/c_{\min}\}$ and $\beta = B/c_{\min}$. Applying the C-ETC framework of Nie et al. (2023) with normalized rewards $\tilde{f}(S) = \frac{1}{U} f(S)$ yields the bound $O(U \cdot (\frac{1}{2} + \tilde{K} + 2\beta)^{2/3} (\tilde{K} K)^{1/3} T^{2/3} \log(T)^{1/3})$. To obtain our clean bound, we use $(\frac{1}{2} + \tilde{K} + 2\beta)^{2/3} \leq (K + \beta)^{2/3}$ and $(\tilde{K} K)^{1/3} \leq K^{2/3}$, then apply $(K + \beta)^{2/3} \cdot K^{2/3} \leq (K + \beta)^{4/3}$ to get the stated bound.

The runtime analysis stems directly from the algorithm of Nie et al. (2023). $\square$

## C  PROOF OF THEOREM 3

**Theorem 3.** *For any algorithm, the worst-case regret over SP instances is $\Omega(U\sqrt{TK})$, and $\Omega(\max\{U\sqrt{T}, \sqrt{TK}\})$ when the SP order and user peaks are known.*

**Proof of Theorem 3**. We prove the lower bounds by constructing specific hard instances for each regime.

**Case 1: Known Order and Peaks** ($R_T = \Omega(U\sqrt{T})$). We construct a family of hard instances parameterized by $\omega = \left((\mu_i, \nu_i)\right)_{i=1}^U \in [1/4, 3/4]^{2U}$. To illustrate the construction, consider a "base case" consisting of three users and four arms with a budget $B = 3$ and costs $c = (1, 3, 1, 1)$. The reward structure for this case is shown in Table 1.

The full instance comprises $U$ copies of User 1 (indexed $1, \ldots, U$) and $2U$ copies of Users 2 and 3 (indexed $U + 1, \ldots, 3U$). The instance is PSP with order $1 \prec 2 \prec 3 \prec 4$. Users $1 \ldots U$ peak at Arm 2, while the others peak at Arm 4. Due to the budget $B = 3$, any feasible matching must either select Arm 2 alone (cost 3), or a subset of $\{1, 3, 4\}$. The optimal matching selects $\{1, 3, 4\}$, assigning users $1 \ldots U$ to $\arg\max(\mu_i, \nu_i)$ and the rest to Arm 4. The optimal expected reward is $V^\star(\omega) = \sum_{i=1}^{U} \max\{\mu_i, \nu_i\} + 2U$.

Consider an algorithm $\mathcal{A}$ generating matchings $\pi^t$. Let $T_2 = \{t \in [T] : \text{Im}(\pi^t) = \{2\}\}$ be the rounds where Arm 2 is selected, and $T_1 = [T] \setminus T_2$. In rounds $t \in T_2$, the reward is exactly $U$. In rounds $t \in T_1$, users $U + 1 \ldots 3U$ contribute at most $2U$ (if they are being matched to Arm 4), while user $i \in [U]$ contributes based on their assignment to 1,3,4. Decomposing the cumulative reward, similar to the classical bandit analysis, we can upper-bound the reward by

$$\mathbb{E}_\omega^{\mathcal{A}} \left[ \sum_{t=1}^{T} V(\pi^t; \Theta^\omega) \right] \leq \mathbb{E}_\omega^{\mathcal{A}} \left[ U|T_2| + 2U|T_1| + \sum_{i=1}^{U} \left( \mu_i N_1^i + \nu_i N_3^i \right) \right],$$

where $N_j^i$ is the number of times user $i$ is matched to arm $j$. The regret is then bounded by:

$$R_T \geq \mathbb{E}_\omega^{\mathcal{A}} \left[ U|T_2| + |T_1| \sum_{i=1}^{U} \max\{\mu_i, \nu_i\} - \sum_{i=1}^{U} \left( \mu_i N_1^i + \nu_i N_3^i \right) \right].$$

If $\mathbb{E}[|T_2|] \geq \sqrt{T}$, the first term yields $\Omega(U\sqrt{T})$. If $\mathbb{E}[|T_2|] < \sqrt{T}$, then $\mathbb{E}[|T_1|] = \Omega(T)$, and the remaining terms correspond to the regret of $U$ independent 2-armed bandit problems played for $\Omega(T)$ rounds. By the standard minimax lower bound (Auer et al., 2002), this scales as $\Omega(U\sqrt{T})$.

**Case 2: Known Order and Peaks ($R_T = \Omega(\sqrt{TK})$).** To capture the dependency on $K$, assume $U = 2$, budget $B = 1$ (forcing a choice of exactly one arm per round), and costs $c_k = 1$. We construct a family of hard PSP instances parameterized by $i^\star \in [K]$. For each $i^\star$, the reward structure is defined as follows: User 1 has preferences $1/2 + \Delta$ for arms $k \leq i^\star$ and $1/2$ for arms $k > i^\star$, while User 2 has preferences $1/2$ for arms $k < i^\star$ and $1/2 + \Delta$ for arms $k \geq i^\star$.

Each instance is PSP: User 1's preferences are non-increasing with a peak at arm 1, and User 2's preferences are non-decreasing with a peak at arm $K$. For any choice of $i^\star$, arm $i^\star$ is the unique optimal arm with total reward $1 + 2\Delta$, while all other arms yield $1 + \Delta$. Identifying the optimal arm among $K$ possibilities reduces to the classical $K$-armed bandit problem. Following the standard change-of-measure argument via Bretagnolle-Huber (e.g., Lattimore & Szepesvári (2020, Chapter 15)) with $\Delta \asymp \sqrt{K/T}$, we obtain a lower bound of $\Omega(\sqrt{TK})$. Note that although each round yields two reward observations rather than one, this only affects the KL-divergence by a constant factor and does not change the order of the lower bound. Combining this with Case 1, we have a regret lower bound of $\Omega(\max\{U\sqrt{T}, \sqrt{TK}\})$.

**Case 3: Unknown Peaks ($R_T = \Omega(U\sqrt{TK})$).** If the peaks are unknown, we rely on the classic construction of the multi-armed bandit lower bound (e.g., Slivkins (2019, Chapter 2)). Specifically, we construct an instance where each user $u$ has a unique, unknown peak $p(u)$ with reward $1/2 + \Delta$, while all other arms offer reward $1/2$. Observe that such preference profiles are trivially PSP regardless of the underlying order of arms, as they are constant everywhere except at a single point (the peak). Since the distributions are independent across users, learning the peak for one user provides no information about the others. This reduces to $U$ independent $K$-armed bandit instances. By the standard minimax lower bound, each user contributes $\Omega(\sqrt{TK})$ to the regret, resulting in a total regret of $\Omega(U\sqrt{TK})$. $\qquad \square$

# D  DEFERRED PROOFS FROM SECTION 4

**Lemma 4.** *Fix any PSP matrix $\Theta$ with peaks $p(\cdot)$, and any arm subset $S = \{k_1, \ldots, k_m\} \subseteq K$ with $k_1 < \cdots < k_m$. Let $\pi^\star \in \arg\max_{\pi, \, \text{Im}(\pi) \subseteq S} V(\pi; \Theta)$. For any user $u$, if $k_j \leq p(u) \leq k_{j+1}$ for some $j$ with $1 \leq j < m$, then $\pi^\star(u) \in \{k_j, k_{j+1}\}$; otherwise, $\pi^\star(u) \in \{k_1, k_m\}$.*

---

**Algorithm 6** SP-MATCHING

---

**Require:** PSP matrix $P$ with peaks $p(\cdot)$, budget $B$, costs $c(\cdot)$
**Ensure:** $\pi^\star = \arg\max_{\pi \in \Pi} V(\pi; P)$
1: Add arm 0 with cost $c_0 = 0$ and $P_{u,0} = 0$ for all $u \in [U]$
2: $\forall i, j \in \{0\} \cup [K] : G_{i,j} \leftarrow \sum_{u:i<p(u)\leq j} \max\{P_{u,i}, P_{u,j}\}$
3: $\forall b = 1, ..., B : F(0, b) \leftarrow 0$
4: **for** $k = 1, \ldots, K$ **do**
5:     **for** $b = c_k, \ldots, B$ **do**
6:         $F(k, b) \leftarrow \max_{\substack{i:0\leq i<k, \\ b\geq c_i+c_k}} [F(i, b - c_k) + G_{i,k}]$
7: $V^\star \leftarrow \max_{k\in[K]} \left\{ F(k, B) + \sum_{u:p(u)>k} P_{u,k} \right\}$
8: Backtrack to find selected arms $S^\star$
9: **return** $\pi^\star(u) = \arg\max_{k\in S^\star} P_{u,k}$

---

**Proof of Lemma 4.** Recall that under single-peaked preferences, the preferences of each user $u$ are unimodal with a peak at $p(u)$—the reward function $\theta_{u,k}$ is non-decreasing for $k \preceq p(u)$ and non-increasing for $p(u) \preceq k$.

For the first case, assume that $k_j \preceq p(u) \preceq k_{j+1}$ for some $1 \leq j < m$. For any $\ell < j$, since $k_\ell \preceq k_j \preceq p(u)$, the non-decreasing property gives $\theta_{u,k_\ell} \leq \theta_{u,k_j}$. Similarly, for any $\ell > j+1$, since $p(u) \preceq k_{j+1} \preceq k_\ell$, the non-increasing property yields $\theta_{u,k_{j+1}} \geq \theta_{u,k_\ell}$. Therefore, any arm outside $\{k_j, k_{j+1}\}$ is dominated by at least one arm in this pair, establishing that $\pi^\star(u) \in \{k_j, k_{j+1}\}$.

For the second case, start with the case of $p(u) \preceq k_1$, which implies $p(u) \preceq k_1 \preceq \cdots \preceq k_m$. By the non-increasing property after the peak, we have $\theta_{u,k_1} \geq \theta_{u,k_2} \geq \cdots \geq \theta_{u,k_m}$, which implies $\pi^\star(u) = k_1$. The case of $k_m \preceq p(u)$ is similar, and yields $\pi^\star(u) = k_m$. $\qquad\square$

**Theorem 5.** *For any PSP matrix,* SP-MATCHING *finds an optimal matching in time* $O(K^2(U+B))$.

**Proof of Theorem 5.** Define $\text{OPT}(k, b)$ as the maximum reward achievable when arm $k$ is the right-most selected arm, the total budget is at most $b$, and we consider only users whose peaks lie in $\{0, 1, \ldots, k\}$. Note that the fictive arm 0 contributes zero reward and zero cost. We prove by induction that $F(k, b) = \text{OPT}(k, b)$ for all $k \geq 0$ and $b \geq 0$.

*Base case:* For the fictive arm, $\text{OPT}(0, b) = 0$ for all $b \leq B$ since no users have peaks at arm 0 and the arm contributes zero reward. The algorithm correctly assigns $F(0, b) = 0$ in Line (3).

*Inductive step:* Assume $F(i, b') = \text{OPT}(i, b')$ for all $0 \leq i < k$ and all budgets $c_k \leq b' < b$. For any $k \geq 1$ and budget $b \geq c_k$, the optimal solution must select some arm $i$ with $0 \leq i < k$ as the second-rightmost selected arm (where $i = 0$ corresponds to selecting only arm $k$).

By Lemma 4, users with peaks in the interval $(i, k]$ are optimally assigned to either arm $i$ or arm $k$, contributing exactly $G_{i,k}$ to the total reward. Users with peaks in $\{0, 1, \ldots, i\}$ contribute optimally according to $\text{OPT}(i, b - c_k)$ by definition. Since the fictive arm 0 has zero cost, the constraint $b \geq c_i + c_k$ is always satisfiable with $i = 0$. Therefore,

$$\text{OPT}(k, b) = \max_{\substack{i:0\leq i<k, \\ b\geq c_i+c_k}} [\text{OPT}(i, b - c_k) + G_{i,k}].$$

The inductive hypothesis ensures that $\text{OPT}(i, b - c_k) = F(i, b - c_k)$, so:

$$\text{OPT}(k, b) = \max_{\substack{i:0\leq i<k, \\ b\geq c_i+c_k}} [F(i, b - c_k) + G_{i,k}] = F(k, b).$$

After computing $F(k, b)$ for all relevant values, Line (7) computes

$$\max_{k\in[K]} \{F(k, B) + \sum_{u:p(u)>k} P_{u,k}\},$$

which accounts for users whose peaks exceed the rightmost selected arm due to the second case in Lemma 4. This ensures that the algorithm correctly computes the maximum reward for the given budget $B$. The backtracking reconstructs the optimal matching, from which we can exclude arm 0 if it was somehow included by picking any other arm for that user.

**Runtime Analysis:** Computing the $G$ matrix requires $O(K^2 U)$ time. The dynamic programming fills $O(KB)$ entries, with each entry requiring $O(K)$ operations, yielding $O(K^2 B)$ time. The remaining steps take $O(KU)$ time. Therefore, the total complexity is $O(K^2 U + K^2 B)$. □

# E  DEFERRED PROOFS AND DETAILS FROM SECTION 5

**Lemma 6.** *For any non-empty confidence set $\mathcal{C}^t(\prec, p, H_t)$, there exists a unique element-wise maximal matrix $P^t \in \mathcal{C}^t$ such that $P_{u,k}^t \geq P_{u,k}$ for all $P \in \mathcal{C}^t$ and all $u \in [U], k \in [K]$. Furthermore, this matrix is given by* $P_{u,k}^t = \begin{cases} \min_{i:k \preceq i \preceq p(u)} \mathrm{UCB}_{u,i}(t), & k \preceq p(u) \\ \min_{i:p(u) \preceq i \preceq k} \mathrm{UCB}_{u,i}(t), & p(u) \preceq k \end{cases}.$

**Proof of Lemma 6.** First, observe that $P^t$ is SP by construction (as the values are defined via a running minimum moving away from the peak) and satisfies $P^t \leq \mathrm{UCB}$ by definition.

Next, assume towards contradiction that there exists a matrix $P \in \mathcal{C}$ and a pair $(u, k)$ such that $P_{u,k} > P_{u,k}^t$. By the definition of $P^t$, there exists an index $k'$ with either $k \preceq k' \preceq p(u)$ or $p(u) \preceq k' \preceq k$ such that $P_{u,k}^t = \mathrm{UCB}_{u,k'}(t)$. From our assumption, we have $P_{u,k} > \mathrm{UCB}_{u,k'}$. Since $P \in \mathcal{C}$, Definition 3 ensures that $P_{u,k'} \leq \mathrm{UCB}_{u,k'}(t)$. Combining these inequalities yields $P_{u,k} > \mathrm{UCB}_{u,k'}(t) \geq P_{u,k'}$. However, the SP property of $P$ with peak at $p(u)$ implies $P_{u,k} \leq P_{u,k'}$ (since both arms lie on the same side of the peak), contradicting $P_{u,k} > P_{u,k'}$.

Finally, if $\mathcal{C}^t$ is non-empty, then for any $P \in \mathcal{C}^t$, we have $P^t \geq P \geq \mathrm{LCB}$, ensuring $P^t \in \mathcal{C}^t$ and completing the proof. □

**Theorem 7.** *For any SP instance with known SP order $\prec$ and peaks $p(\cdot)$, MVM achieves regret of at most $O(U\sqrt{TK \ln T})$ with per-round runtime of $O(K^2 U + K^2 B)$.*

**Proof of Theorem 7.** The analysis follows the same structure as the UCB-based algorithm in Corollary 2, with the maximal matrix $P^t$ playing the role of the UCB matrix. We condition on the clean event $\mathcal{E} = \{\Theta \in \mathcal{C}^t \text{ for all } t\}$, which holds with probability at least $1 - {}^{2UK}/_{T^3}$ by Hoeffding's inequality and a union bound.

Under $\mathcal{E}$, since $P^t$ is element-wise maximal in $\mathcal{C}^t$ (Lemma 6), the per-round regret satisfies Inequality (3). Summing over rounds and applying the same regrouping and Jensen's inequality arguments as in the proof of Corollary 2 yields $R_T = O(U\sqrt{TK \ln T})$.

For the runtime, each round involves updating statistics ($O(U)$), constructing $P^t$ via Lemma 6 ($O(UK)$), and running SP-MATCHING ($O(K^2(U + B))$) by Theorem 5). The complexity is dominated by the matching step. □

# F  DEFERRED PROOFS FROM SECTION 6

**Proposition 8.** *A matrix is SP w.r.t. an order $\prec$ if and only if it is 0-ASP w.r.t. $\prec$.*

**Proof of Proposition 8.** First, assume $P$ is SP w.r.t. $\prec$. By definition, for any user $u$, the values $P_{u,k}$ are non-decreasing up to a peak $p(u)$ and non-increasing thereafter. Consequently, for any triplet $i \prec j \prec \ell$, the middle element $j$ must satisfy $P_{u,j} \geq P_{u,i}$ (if $j \preceq p(u)$) or $P_{u,j} \geq P_{u,\ell}$ (if $j \succeq p(u)$). In either case, $P_{u,j} \geq \min\{P_{u,i}, P_{u,\ell}\}$, satisfying the 0-ASP condition.

Conversely, suppose $P$ is 0-ASP w.r.t. $\prec$. For any user $u$, let $p(u)$ be an index maximizing $P_{u,\cdot}$. For any $i \prec j \prec p(u)$, the 0-ASP condition implies $P_{u,j} \geq \min\{P_{u,i}, P_{u,p(u)}\} = P_{u,i}$, ensuring non-decreasing values to the left of the peak. By symmetry, values are non-increasing to the right of the peak, implying that $P$ is SP. □

**Lemma 9.** *Let $\tilde{P}$ be a $\delta$-ASP matrix w.r.t. $\prec$. There exists a matrix $P$ which is SP w.r.t. $\prec$, and satisfies $\|P - \tilde{P}\|_\infty \leq \delta$.*

**Proof of Lemma 9**. The proof is via construction. For every $u \in U$, set $p(u) = \arg\max_{k \in [K]} \tilde{P}_{u,k}$, and define $P$ as follows:

$$P_{u,k} = \begin{cases} \max_{i:i \preceq k} \tilde{P}_{u,i}, & k \preceq p(u) \\ \max_{i:k \preceq i} \tilde{P}_{u,i}, & p(u) \preceq k \end{cases}.$$

The matrix $P$ is SP w.r.t. $\prec$ by construction: for each user $u$, the row $P_{u,(\cdot)}$ is defined as a running maximum from the endpoints toward $p(u)$, ensuring non-decreasing values for $k \preceq p(u)$ and non-increasing values for $k \succeq p(u)$.

It remains to show that $\|P - \tilde{P}\|_\infty \leq \delta$. Consider an entry $(u, k)$ with $k \prec p(u)$. By construction, $P_{u,k} = \max_{i:i \preceq k} \tilde{P}_{u,i}$. If the maximum is achieved at $k$ itself, then $P_{u,k} = \tilde{P}_{u,k}$. Otherwise, the maximum is achieved at some $i \prec k$, and we have $P_{u,k} = \tilde{P}_{u,i} \leq \tilde{P}_{u,k} + \delta$, where the inequality follows from applying Definition 4 to the triplet $i \prec k \prec p(u)$. A symmetric argument applies when $p(u) \prec k$. Since $P_{u,k} \geq \tilde{P}_{u,k}$ by construction (as the maximum includes $\tilde{P}_{u,k}$ itself), we conclude $|P_{u,k} - \tilde{P}_{u,k}| \leq \delta$ for all $(u, k)$. $\qquad\square$

**Lemma 10.** *Let $\tilde{P}$ be a matrix such that $\|\tilde{P} - P\|_\infty \leq \varepsilon$ for some SP matrix $P$. EXTRACT-ORDER$(\tilde{P}, \varepsilon)$ returns an order $\prec$ such that $\tilde{P}$ is $(2K\varepsilon)$-ASP w.r.t. $\prec$.*

**Proof of Lemma 10**. We first establish the existence of an order that satisfies all the constraints; we do so via the SP order of $P$, which we denote as $\prec_P$. Fix some $u \in U$ for which a contiguity constraint on the set of arms $S = \{k_1^u, \dots, k_m^u\}$ was imposed. Due to Lines 3–4, we must have $\min_{k \in S} \tilde{P}_{u,k} - \max_{k' \notin S} \tilde{P}_{u,k'} > 2\varepsilon$. Since $\|\tilde{P} - P\|_\infty \leq \varepsilon$, this implies $\min_{k \in S} P_{u,k} - \max_{k' \notin S} P_{u,k'} > 0$. Thus, because $P$ is 0-ASP w.r.t. $\prec_P$, $\prec_P$ must satisfy this constraint as well; otherwise, as was previously discussed, we could have found a triplet of arms that contradicts Definition 4.

Next, let $\prec$ be a returned order (one must exist, as our first step suggests). Fix $u$ and arms $i \prec j \prec \ell$. We must show $\tilde{P}_{u,j} \geq \min\{\tilde{P}_{u,i}, \tilde{P}_{u,\ell}\} - 2K\varepsilon$. Assume w.l.o.g. that $\tilde{P}_{u,i} \leq \tilde{P}_{u,\ell}$. Consider the arms sorted by preference $k_1^u, \dots, k_K^u$. If there were any index $m$ such that $\{k_1^u, \dots, k_m^u\}$ includes $i$ and $\ell$ but excludes $j$, and the gap $\tilde{P}_{u,k_m^u} - \tilde{P}_{u,k_{m+1}^u} > 2\varepsilon$, the algorithm would constrain this set to be contiguous. This would force $j$ outside the interval between $i$ and $\ell$, contradicting $i \prec j \prec \ell$. Therefore, no gap exceeding $2\varepsilon$ exists in the sorted sequence between the values $\tilde{P}_{u,i}$ and $\tilde{P}_{u,j}$. Summing these gaps (at most $K$) yields $\tilde{P}_{u,i} - \tilde{P}_{u,j} \leq 2K\varepsilon$. $\qquad\square$

**Theorem 11.** EMC$\left(N = \left\lceil T^{2/3}(\ln T)^{1/3}\right\rceil\right)$ *yields an expected regret of at most $\tilde{O}(UKT^{2/3})$.*

**Proof of Theorem 11**. Let $\varepsilon = \sqrt{2\ln T/N}$. By Hoeffding's inequality and a union bound, the clean event $\mathcal{E} = \{\|\bar{\Theta} - \Theta\|_\infty \leq \varepsilon\}$ holds with probability at least $1 - 2KU/T^4$. Its complement, $\mathcal{E}^C$, occurs with low probability and contributes at most $TU$ to the overall regret; thus, we condition the rest of the proof on $\mathcal{E}$. Under $\mathcal{E}$, the approximation guarantees from Lemmas 9 and 10 combined imply that the constructed matrix $\tilde{\Theta}$ satisfies $\|\tilde{\Theta} - \bar{\Theta}\|_\infty \leq 2K\varepsilon$. By the triangle inequality, the total estimation error is bounded by

$$\|\tilde{\Theta} - \Theta\|_\infty \leq (2K+1)\sqrt{2\ln T/N}. \tag{6}$$

Now, let $\tilde{\pi}$ be the matching computed in Line (5), and let $\pi^\star$ be the optimal matching w.r.t. $\Theta$. Using Inequality (6), we can bound the regret of a single commitment round ($t > KN$) as follows:

$$V(\pi^\star; \Theta) - V(\tilde{\pi}; \Theta) \leq V(\pi^\star; \tilde{\Theta}) - V(\tilde{\pi}; \tilde{\Theta}) + 2U(2K+1)\sqrt{2\ln T/N} \leq 2U(2K+1)\sqrt{2\ln T/N},$$

where the last inequality follows the optimality of $\tilde{\pi}$ with respect to $\tilde{\Theta}$. Thus, by dividing into exploration and commitment rounds, we can bound the total regret by

$$R_T \leq O\left(KNU + T \cdot 2U(2K+1)\sqrt{2\ln T/N}\right).$$

Substituting $N = \lceil T^{2/3}(\ln T)^{1/3} \rceil$ yields $R_T \leq O(UKT^{2/3}(\ln T)^{1/3}) = \tilde{O}(UKT^{2/3})$, which completes the proof. $\qquad\square$

**Theorem 12.** *Consider the following optimization problem, termed* MAX-SP-WCS*: given sets of users $U$ and arms $K$, a matching $\pi : U \to K$, and confidence intervals $[\text{LCB}_{u,k}, \text{UCB}_{u,k}]$ for each $(u,k) \in U \times K$, find $\max_{P \in \mathcal{C}} V(\pi; P)$, where $\mathcal{C}$ contains all SP matrices respecting the confidence intervals. Then, approximating* MAX-SP-WCS *within a factor of $\frac{3}{4} + \delta$ is NP-hard for any $\delta > 0$.*

**Proof of Theorem 12.** We reduce from MAX BETWEENNESS (Opatrny, 1979; Austrin et al., 2015), which involves, given a set of elements $S$ and a collection of ordered triplets $G \subseteq S^3$, finding a linear order of $S$ that maximizes the number of triplets $(a, b, c) \in G$ for which element $b$ lies between $a$ and $c$ in the order. Austrin et al. (2015) show that for any $\eta > 0$, it is NP-hard to distinguish between instances where at least $(1 - \eta)$ fraction of triplets can be satisfied versus at most $(1/2 + \eta)$ fraction.

**Reduction.** Given a BETWEENNESS instance $(S, G)$, construct a MAX-SP-WCS instance as follows. Create an arm for each element $s \in S$, so $K = S$. For each triplet $g = (a_g, b_g, c_g) \in G$, create two users $u_g$ and $v_g$, so $U = \{u_g, v_g : g \in G\}$. Define the matching by $\pi(u_g) = a_g$ and $\pi(v_g) = c_g$. Fix $\varepsilon \in (0, 1/2)$ and set confidence intervals as:

$$[\text{LCB}_{u_g,k}, \text{UCB}_{u_g,k}] = \begin{cases} [1 - \varepsilon, 1], & k = b_g \\ [0, \varepsilon], & k = c_g \\ [0, 1], & \text{otherwise} \end{cases},$$

$$[\text{LCB}_{v_g,k}, \text{UCB}_{v_g,k}] = \begin{cases} [1 - \varepsilon, 1], & k = b_g \\ [0, \varepsilon], & k = a_g \\ [0, 1], & \text{otherwise} \end{cases}.$$

This construction is polynomial in the size of the BETWEENNESS instance.

**Key observation.** Fix any arm order $\prec$ and triplet $g = (a_g, b_g, c_g)$. If $b_g$ lies between $a_g$ and $c_g$ in $\prec$, we can complete both rows $u_g$ and $v_g$ unimodally with peaks at $b_g$ that respect the confidence intervals and achieve $P_{u_g,a_g} = P_{v_g,c_g} = 1$: set $P_{u_g,k} = 1$ for all arms $k$ on the same side of $b_g$ as $a_g$ (including $b_g$), and $P_{u_g,k} = 0$ otherwise; similarly for $v_g$.

Conversely, if $b_g$ does not lie between $a_g$ and $c_g$, then either $c_g$ lies between $a_g$ and $b_g$, or $a_g$ lies between $b_g$ and $c_g$. In the former case, single-peakedness requires $P_{u_g,a_g} \leq P_{u_g,c_g}$ or $P_{u_g,b_g} \leq P_{u_g,c_g}$; since the confidence intervals enforce $P_{u_g,c_g} \leq \varepsilon$ and $P_{u_g,b_g} \geq 1 - \varepsilon$, we must have $P_{u_g,a_g} \leq \varepsilon$. By symmetry, in the latter case, $P_{v_g,c_g} \leq \varepsilon$. These bounds can be achieved with equality.

Therefore, the contribution from users $u_g, v_g$ is exactly 2 if triplet $t$ is satisfied by $\prec$, and $1 + \varepsilon$ otherwise. Hence, if $s$ triplets are satisfied:

$$V(\pi; P) = 2s + (|G| - s)(1 + \varepsilon) = |G|(1 + \varepsilon) + s(1 - \varepsilon).$$

**Hardness of approximation.** Consider a MAX BETWEENNESS instance $I$ that is hard to distinguish. Denote the corresponding MAX-SP-WCS instance by $\hat{I}$, and let $\text{OPT}(\hat{I})$ denote the optimal value of $\hat{I}$. If at least $(1 - \eta)|G|$ triplets of $I$ can be satisfied, which we refer to as case $H$, we would have:

$$\text{OPT}(\hat{I}_H) \geq |G|(1 + \varepsilon) + |G|(1 - \eta)(1 - \varepsilon) = |G|(2 - \eta + \varepsilon\eta).$$

On the other hand, if at most $(1/2 + \eta)|G|$ triplets can be satisfied, which we refer to as case $L$:

$$\text{OPT}(\hat{I}_L) \leq |G|(1 + \varepsilon) + |G|(1/2 + \eta)(1 - \varepsilon) = |G|\left(3/2 + \eta + \varepsilon/2 - \varepsilon\eta\right).$$

Suppose there exists an $\alpha$-approximation algorithm ALG for MAX-SP-WCS, i.e., $\text{ALG}(\tilde{I}) \geq \alpha \cdot \text{OPT}(\tilde{I})$ for all MAX-SP-WCS instances $\tilde{I}$. Then, in case $H$, we would have $\text{ALG}(\hat{I}_H) \geq \alpha \cdot \text{OPT}(\hat{I}_H)$ while in case $L$ $\text{ALG}(\hat{I}_L) \leq \text{OPT}(\hat{I}_L)$ will hold trivially. Notice that if $\text{ALG}(\hat{I}_H) > \text{OPT}(\hat{I}_L)$, we could distinguish the two cases. This holds whenever:

$$\alpha > \frac{\text{OPT}(\hat{I}_L)}{\text{OPT}(\hat{I}_H)} \leq \frac{\frac{3}{2} + \eta + \frac{\varepsilon}{2} - \varepsilon\eta}{2 - \eta + \varepsilon\eta} \xrightarrow{\eta, \varepsilon \to 0} \frac{3/2}{2} = \frac{3}{4}.$$

Thus, for any $\delta > 0$, a $(\frac{3}{4} + \delta)$-approximation algorithm would distinguish between hard instances for sufficiently small $\eta, \varepsilon$, contradicting the NP-hardness of MAX BETWEENNESS. $\qquad\square$

## G   HANDLING TIES IN PEAK INDICES

In the classical study of single-peaked preferences in social choice theory, preferences are ordinal—each user provides a ranking over alternatives rather than cardinal utility values. In this ordinal setting, ties at the peak do not arise: the peak is simply the top-ranked alternative, which is unique by definition. However, in our cardinal framework, a user $u$ may have multiple indices $k$ that achieve the same maximal expected reward $\max_{k' \in [K]} \theta_{u,k'}$. When this occurs, the peak $p(u)$ as defined in Definition 1 is not unique, and we must verify that our algorithms and analyses are robust to arbitrary tie-breaking choices.

**Robustness of the Offline Algorithm.**   We begin by establishing that Lemma 4 is robust to the arbitrary choice of peak when a user has several maximizers. Suppose user $u$ has two peak indices $x < y$ with $\theta_{u,x} = \theta_{u,y} = \max_{k \in [K]} \theta_{u,k}$, and fix a selected set $S = \{k_1 < k_2 < \cdots < k_m\}$. We claim that the value of the optimal matching of $u$ against $S$ is independent of whether we set $p(u) = x$ or $p(u) = y$. To see this, observe that exactly one of the following cases holds:

- There exists $j \in \{1, \ldots, m-1\}$ with $k_j \le x \le y \le k_{j+1}$ and no $k_\ell \in (x, y)$. By unimodality, any arm outside $\{k_j, k_{j+1}\}$ is dominated by one of them, so $\pi^\star(u) \in \{k_j, k_{j+1}\}$ regardless of the tie-break.

- $y \le k_1$. Values are non-increasing to the right of the peaks, hence $\pi^\star(u) = k_1$ regardless of the tie-break.

- $k_m \le x$. Values are non-decreasing to the left of the peaks, hence $\pi^\star(u) = k_m$ regardless of the tie-break.

- There exist indices $i \le j$ with $x \le k_i \le \cdots \le k_j \le y$. Since all $k \in [x, y]$ attain the peak value by the single-peaked property, any $k \in \{k_i, \ldots, k_j\}$ is optimal. The lemma asserts $\pi^\star(u) \in \{k_{i-1}, k_i\}$ if the peak is set to $x$, and $\pi^\star(u) \in \{k_j, k_{j+1}\}$ if the peak is set to $y$. In either case, the attained value $\max_{k \in S} \theta_{u,k}$ remains unchanged.

In every case, user $u$'s contribution to the matching value is invariant to the choice of peak.

This invariance extends to SP-MATCHING (Algorithm 6). Observe that every choice of the rightmost arm $k$ in Line 7 corresponds to a subset of selected arms and a matching that assigns each user according to Lemma 4. By the argument above, the attained value of each user in this subset is independent of how we resolve peak ties. The choice of peak only affects *when* we account for a user's contribution during the dynamic programming computation, not the contribution itself. Therefore, running SP-MATCHING with different valid peak indices yields the same optimal value $V^\star$.

**Robustness of the Online Algorithm with Known Structure.**   For the MVM algorithm (Algorithm 1), we must verify that the maximal matrix construction in Lemma 6 remains valid when users have multiple peak indices. Suppose user $u$ has peak indices $p, p+1, \ldots, p+m$ for some $m \ge 0$, and this information is known to the algorithm. The maximal matrix consistent with this structure should assign the same value to all peak indices, equal to the minimum of their upper confidence bounds. Formally, one could define:

$$
P_{u,k}^t = \begin{cases} \min_{i \in \{0,\ldots,m\}} \mathrm{UCB}_{u,p+i}(t), & k \in \{p, p+1, \ldots, p+m\} \\ \min_{i:k \preceq i \preceq p+m} \mathrm{UCB}_{u,i}(t), & k \prec p \\ \min_{i:p \preceq i \preceq k} \mathrm{UCB}_{u,i}(t), & p+m \prec k. \end{cases}
$$

However, for any choice of peak index $p + i$ with $i \in \{0, \ldots, m\}$, using the original definition from Lemma 6 yields a matrix that element-wise upper bounds this modified construction. Thus, the regret analysis of Theorem 7 remains valid even if we use a slightly loose upper bound that may technically fall outside the confidence set. The key properties used in the analysis—that $P^t$ upper bounds any matrix in the confidence set and that $P^t$ is single-peaked with respect to $\prec$—continue to hold.

**Unknown Structure Regime.** For the algorithms in Section 6, peak ties do not require special treatment. In this regime, peaks are not explicitly computed or used by the algorithm; they only appear implicitly through the call to SP-MATCHING in Line 5 of Algorithm 3. Since we have already established that SP-MATCHING is invariant to the choice of peak indices, the analysis of EMC and its extensions remains valid regardless of how ties are resolved.

# H    EXPERIMENTAL VALIDATION

In this section, we present computational experiments to demonstrate the practicality of our algorithms and provide empirical validation of the theoretical regret bounds. [2]

## H.1    SIMULATION DETAILS

We now describe the synthetic setup used to evaluate our algorithms, including instance generation, implementation details, and experimental protocol.

**Instance generation.** We constructed single-peaked expected reward matrices $\Theta$ using the following procedure: for each user $u$, we first sampled $K$ preference values independently from a uniform distribution over $[0.2, 0.9]$. We then selected a random peak location $p(u) \in [K]$ and arranged the values to create a unimodal preference profile—The largest sampled value was assigned to the peak, while the remaining values were sorted and distributed to create an increasing sequence up to the peak and a decreasing sequence afterward.

The actual rewards observed during algorithm execution were drawn from Bernoulli distributions, where $r_{u,k} \sim \mathrm{Ber}(\theta_{u,k})$ for each user-arm pair. We set unit costs for all arms ($c_k = 1$ for all $k$) and used a budget constraint of $B = \lfloor K/2 \rfloor$, allowing selection of approximately half the available arms in each round.

**Algorithmic implementation.** The implementation of EMC and MVM uses standard Python scientific computing libraries (NumPy, Pandas) without GPU acceleration, as the algorithms are primarily CPU-bound. The SP-MATCHING algorithm, used by both algorithms, was implemented using the dynamic programming approach described in Algorithm 6. Additionally, for the PQ tree implementation, we used the SageMath library (The Sage Developers, 2025), a standard Python package for advanced mathematical computations.

**Experimental setup.** We tested both algorithms on 10 random instances with $U = 100$ users and $K = 20$ arms. We estimated the expected regret using 10 independent runs for each instance. For the EMC algorithm, we tested time horizons ranging from $T = 10^5$ to $T = 10^6$ rounds. To simulate the unknown order setting, we randomly permuted the columns of each generated single-peaked matrix before running the algorithm. For the MVM algorithm, we executed a single run up to $T = 10^5$ rounds, recording the cumulative regret at each time step to observe the full regret trajectory. This method naturally serves as an upper bound on the regret that would be incurred if the algorithm were run separately for smaller values of $T$. However, we still observe the same predicted asymptotic behavior. Since this algorithm assumes a known order and peaks, we provided the true single-peaked structure directly without column permutation.

The difference in the number of rounds $T$ for the two algorithms stems from the fact that EMC necessitates larger time horizons to ensure a sufficient number of commitment rounds relative to the exploration rounds.

**Used hardware.** The complete simulation suite was executed on a standard MacBook Pro with 18 GB of RAM and 8 CPU cores. We utilized parallel processing across five cores to accelerate computation. The complete set of experiments required approximately 6 hours of computation time.

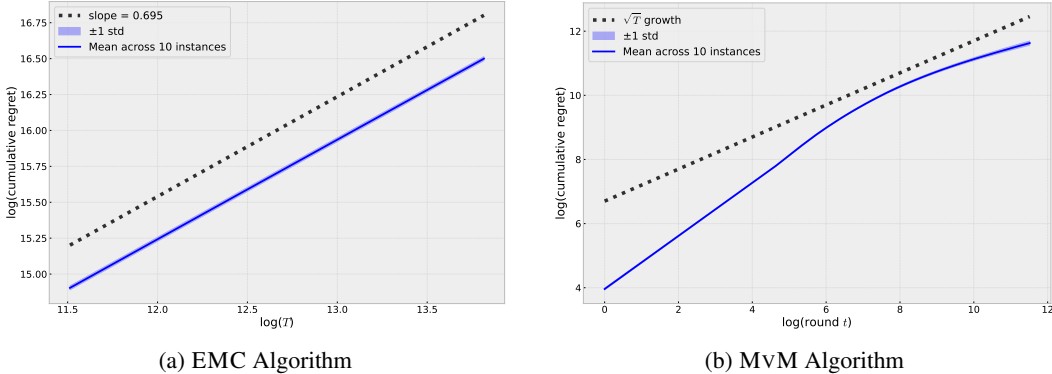

(a) EMC Algorithm  (b) MᴠM Algorithm

Figure 2: Log-log plots of cumulative regret versus time for both algorithms. Each plot shows the mean regret over all 10 instances (with 10 runs each) and shaded regions indicating standard deviation. The EMC algorithm (left) achieves a slope of approximately 0.69, approaching the theoretical guarantee of $2/3 \approx 0.67$. The MᴠM algorithm (right) demonstrates slopes below 0.5, consistent with the theoretical bound.

## H.2 RESULTS

Figure 2 presents log-log plots of cumulative regret versus time for both algorithms. Each plot displays the mean regret computed over all 10 instances (with 10 runs each), along with shaded regions representing the standard deviation. The narrow confidence bands demonstrate the consistency of our results across different problem instances and random seeds. Note that the two plots use different x-axis representations: the EMC plot (Figure 2a) shows $\log(T)$ where each point represents the total regret after $T$ rounds of interaction, while the MᴠM plot (Figure 2b) shows $\log(t)$ with the cumulative regret trajectory at each round $t$.

**The EMC algorithm.** As shown in Figure 2a, the algorithm exhibits consistent behavior, as evidenced by the nearly indistinguishable standard deviation bands. The fitted slope is approximately 0.694, approaching but not quite reaching the theoretical prediction of $2/3 \approx 0.667$ from our $\tilde{O}(UKT^{2/3})$ regret bound. When tested for larger time horizons, the empirical slope approaches the theoretical value of $2/3$, confirming the asymptotic prediction of our analysis.

**The MᴠM algorithm.** As shown in Figure 2b, the algorithm demonstrates excellent empirical performance with slopes ranging from 0.388 to 0.434 across different instances. All observed slopes fall strictly below the theoretical upper bound of 0.5, corresponding to our $\tilde{O}(U\sqrt{KT})$ regret bound. The variation in slopes across instances reflects the algorithm's adaptive nature: instances with more challenging preference structures result in steeper regret growth, which corresponds to slower learning. The small standard deviation bands indicate that the algorithm's performance is stable across multiple runs of the same instance. This gap between empirical performance and theoretical bounds suggests that our analysis may be conservative for typical problem instances.

**Comparison.** The empirical slopes–0.694 for EMC versus 0.388-0.434 for MᴠM–highlight the cost of unknown order. While both algorithms achieve sublinear regret, the MᴠM algorithm's knowledge of the single-peaked structure enables significantly better asymptotic performance. The EMC algorithm's performance is hindered by the finite-horizon effects discussed above, but still achieves sublinear regret that would approach the theoretical rate with larger time horizons.

## I BIPARTITE BUDGETED MATCHING VARIANT

In this appendix, we address a recommendation setting where each arm may serve at most one user.

---

[2]Our code is available at https://github.com/GurKeinan/code-for-Bandits-with-Single-Peaked-Preferences-and-Limited-Resources-paper.

### I.1 PROBLEM FORMULATION

We consider the same stochastic setting as in our main model $(T, U, K, (D_{u,k})_{u,k}, (c_k)_k, B)$, but with a different notion of a valid matching.

In our main model, multiple users may be assigned to the same arm in a round, and the budget constraint applies to the *set* of distinct arms used. In the bipartite variant considered here, each user can be matched to at most one arm, and each arm can be matched to at most one user (one-to-one matching). For simplicity, we focus on the case of unit costs; the case of general costs remains an interesting problem for future work. In each round, the learner chooses a matrix $X \in \{0,1\}^{U \times K}$ satisfying

$$\sum_k x_{u,k} \leq 1 \ \forall u, \qquad \sum_u x_{u,k} \leq 1 \ \forall k, \qquad \sum_{u,k} x_{u,k} \leq B,$$

and receives an expected reward $V(\pi; \Theta) = \sum_{u,k} \Theta_{u,k} x_{u,k}$. Similarly to before, we denote by $\Pi^{\mathrm{Bi}}$ the set of feasible matchings, where each matching corresponds to a matrix that satisfies these constraints. The only change from our main model is the feasible set of matchings; the stochastic structure and regret definition remain unchanged.

### I.2 POLYNOMIAL-TIME ALGORITHM FOR THE OFFLINE PROBLEM

We start our analysis by showing that the offline budgeted bipartite matching problem can be solved in polynomial time using a reduction to either the Hungarian algorithm or a min-cost flow formulation.

**Proposition 18.** *We can solve* $\arg\max_{\pi \in \Pi^{\mathrm{Bi}}} V(\pi; \Theta)$ *in polynomial time.*

**Proof of Proposition 18.** We divide our proof into two cases based on the relation between the budget and the number of users and arms.

**Case 1:** $B \geq \min(U, K)$. In this case, the budget constraint is non-binding, and we can simply find the maximum weight matching in the bipartite graph formed by users and arms. Formally, we first pad the smaller side of the bipartite graph with dummy nodes to make it square. W.l.o.g., assume $U \leq K$; we add $K - U$ dummy users with $\Theta_{u,k} = 0$ for all arms $k$. We then apply the Hungarian algorithm to find the maximum weight matching in this square bipartite graph, which runs in $O(K^3)$ time. Since the padded users do not contribute to the reward, the resulting matching is optimal for our original problem.

**Case 2:** $B < \min(U, K)$. In this case, the budget constraint is binding, so we formulate the problem as a min-cost flow problem. We construct a flow network as follows:

- Create a source node $s$ and a sink node $t$.

- Create a node $A_u$ for each user $u$ and a node $B_k$ for each arm $k$.

- Add edges from $s$ to each user node $A_u$ and from each arm node $B_k$ to $t$ with capacity 1 and cost 0.

- For each user-arm pair $(u, k)$, add an edge from $A_u$ to $B_k$ with capacity 1 and cost $-\Theta_{u,k}$.

- Node $s$ has a supply of $B$, and node $t$ has a demand of $B$.

- Since the total supply at $s$ is $B$ and the total demand at $t$ is $B$, any valid flow of value $B$ corresponds to selecting exactly $B$ edges between users and arms.

By the Integrality Theorem of minimum cost flow, since all arc capacities and node supplies/demands are integers, there exists an integer-valued optimal flow $f^*$. In our network, since capacities are 1, this implies the flow on any edge $(A_u, B_k)$ is either 0 or 1. We construct the matching $X$ by setting $x_{u,k} = 1$ if and only if the flow on edge $(A_u, B_k)$ is 1. The capacity constraints on edges $(s, A_u)$ ensure $\sum_k x_{u,k} \leq 1$, and the capacity constraints on edges $(B_k, t)$ ensure $\sum_u x_{u,k} \leq 1$. Finally, the total flow value of $B$ ensures $\sum_{u,k} x_{u,k} = B$.

The total cost of the flow is $\sum_{u,k} x_{u,k}(-\Theta_{u,k}) = -\sum_{u,k} x_{u,k}\Theta_{u,k}$. Therefore, minimizing the cost is equivalent to maximizing the total reward $V(\pi; \Theta)$. Since min-cost flow can be solved in polynomial time, the offline bipartite budgeted matching problem is efficiently solvable. $\qquad\square$

## I.3 ONLINE ALGORITHM

Since the offline optimization problem can be solved efficiently, we can employ standard Combinatorial MABs algorithms without relying on the single-peaked structure. Although we can adapt the algorithm from Corollary 2 to this setting, we prefer to use the COMBUCB1 algorithm from Kveton et al. (2015a) and its guarantees with our efficient offline solver as the oracle.

**Corollary 19.** *There exists an efficient online algorithm for the bipartite budgeted matching problem that achieves $\tilde{O}(\sqrt{T})$ regret.*

**Proof of Corollary 19**. We can apply COMBUCB1 using the offline solver from Proposition 18 as the offline oracle. Since the reward function $V(\pi; \Theta)$ is linear in the matchings and the offline oracle is exact, the analysis of Kveton et al. (2015a) applies directly. The worst-case regret is bounded by $O(\sqrt{LK_{tot}T \ln T})$, where $L = UK$ is the number of total items (where item in this setting corresponds to choosing to match a user and item), and $K_{tot} = B$ is the maximum number of chosen items in a valid matching. Thus, the regret scales as $\tilde{O}(\sqrt{UKBT})$, and the per-round computational complexity is polynomial in $U, K$, and $B$. $\qquad\square$

## I.4 LOWER BOUND

We complement our algorithmic result with a matching lower bound, showing that the dependency on $U, K$, and $T$ is unavoidable.

**Theorem 20.** *For the bipartite budgeted matching problem with $K \geq U$, any online algorithm incurs an expected regret of $\Omega(\sqrt{UKT})$.*

**Proof of Theorem 20**. Construct a hard instance where the set of arms $[K]$ is partitioned into $U$ disjoint subsets $S_1, \ldots, S_U$, each of size $|S_u| \geq 2$. User $u$ receives rewards only from arms in $S_u$. Specifically, for each user $u$, one arm in $S_u$ yields Bernoulli rewards with mean $1/2 + \epsilon$, while others in $S_u$ have mean $1/2$. Arms outside $S_u$ yield 0. Since the subsets are disjoint and users are only compatible with their specific subsets, the global problem decomposes into $U$ independent multi-armed bandit instances. The constraints $\sum_k x_{u,k} \leq 1$ and $\sum_u x_{u,k} \leq 1$ are satisfied independently within each subset. Assuming sufficient global budget $B \geq U$, the total regret is the sum of regrets from $U$ independent bandits, each with $K/U$ arms over $T$ rounds. Using the standard minimax lower bound of $\Omega(\sqrt{K'T})$ for $K'$ arms:

$$R(T) = \sum_{u=1}^{U} \Omega\left(\sqrt{\frac{K}{U}T}\right) = \Omega\left(U\sqrt{\frac{KT}{U}}\right) = \Omega\left(\sqrt{UKT}\right).$$

$\qquad\square$

