# OpenReview forum: "Bandits with Single-Peaked Preferences and Limited Resources"
_ICLR.cc/2026/Conference — ICLR 2026 Poster_

### Official Review · Reviewer_jqPn · 2025-10-22

**Soundness:** 3
**Presentation:** 3
**Contribution:** 2
**Rating:** 4
**Confidence:** 3

**Summary:**

This work studies an online stochastic matching problem involving $U$ users and $K$ arms, aiming to maximize the cumulative reward over $T$ rounds under budget constraints. Without assuming any structural property of user preferences, the problem is NP-hard, rendering online learning computationally infeasible. To address this, the authors focus on the case of single-peaked preferences. They first propose an efficient algorithm that achieves a regret of $\tilde{O}(UKT^{2/3})$. When the single-peaked structure is known in advance, they further develop an algorithm that attains a regret of $\tilde{O}(U\sqrt{TK})$.

**Strengths:**

This is the first work to study the matching bandit setting under single-peaked preferences, where an efficient algorithm can be designed. The proposed method achieves a tight regret bound when the preference structure is known.

**Weaknesses:**

The main concern lies in the tightness of the regret bound under the unknown structure. The authors propose an ETC-based algorithm that achieves a regret of order $T^{2/3}$, which appears to be suboptimal. Establishing a tight regret bound for the unknown-structure case would be a meaningful goal, as the current result seems incomplete. If achieving a tighter bound is not possible, the authors should at least provide a corresponding lower bound to justify that the $T^{2/3}$ rate is optimal. Moreover, the ETC and UCB algorithms themselves do not appear to be novel.

**Questions:**

Is there a specific reason for using the term ``preference'' here? To me, preference suggests a Bradley--Terry--Luce--type model, where the reward of each arm is defined relatively, depending on the assigned counterpart.

---

> ### Author Response · Authors · 2025-11-16
>
> We thank the reviewer for their evaluation and feedback. Following the reviewer concern, we intend to both add an algorithm for the unknown structure regime that achieves optimal regret under some assumptions, and to add clarification about our lower bounds that might clarify the nature of the gap. We address both this concern and the reviewer’s question below.
>
> **Weakness**: The reviewer raises a concern about "tightness of the regret bound under the unknown structure." We divide our response into two parts:
>
> **New algorithm**: If the entries of the expectation matrix are sufficiently separated, we can devise an *efficient and optimal* algorithm for the unknown structure regime. Formally, there exists a hybrid algorithm for the unknown structure regime that combines EMC and MvM to obtain regret of approximately
> $$
> O \left(\min ( T^{2/3}, \sqrt{T}+\frac{1}{\Delta^2} ) \right),
> \quad
> \Delta=\min_{u} \min_{i \neq j}|\Theta_{u,i}-\Theta_{u,j}|.
> $$
>
> Importantly, this regret bound strictly improves on that of EMC, and, for $\Delta \geq T^{-1/4}$, it matches the lower bound for this regime. This algorithm follows the spirit of Section 5 (Lines 465–471): first, explore entries of the reward matrix in a round-robin fashion until any two entries in the same row can be reliably compared (within $1/\Delta^2$ rounds by concentration). Then use these estimates to extract the SP order and user peaks and run MvM on the inferred structure, which -- with high probability -- matches the true SP structure. If after at most $T^{2/3}$ rounds the confidence intervals are not disjoint, we switch to the matching and commitment protocol of EMC.
>
> **Clarification about the lower bound and the apparent gap.** As we note in Section 2.2 and Appendix F, an inefficient algorithm can achieve the optimal $O(U\sqrt{TK})$ regret even for non-single-peaked instances. Thus, the challenge in improving regret is *computational*, not *statistical*. Our lower bounds (Theorem 9) are intentionally information-theoretic: they show that SP instances remain statistically non-trivial, since even with the SP structure every algorithm must incur $\Omega(U\sqrt{TK})$ regret. However, these bounds do not incorporate computational constraints and therefore do not match the $O(UKT^{2/3})$ regret achieved by our *efficient* algorithm. Closing this gap would require a fundamentally different type of result, which can be informally stated as “achieving $o(T^{2/3})$ regret in SP instances is NP-hard”. This is a rare and technically challenging type of result in bandit theory. We will clarify in the revision that this mismatch reflects computational, not statistical, limitations.
>
> Moreover, the reviewer stated that "the ETC and UCB algorithms themselves do not appear to be novel." While we agree that the general ideas of Explore-Then-Commit and UCB have been widely studied, our contribution lies in adapting these classical principles to a novel setting, which necessitated the use of new techniques and transformations to overcome the inherent hardness of the offline problem. Specifically, in the unknown structure regime, we indeed employ an exploration and commitment phase, but to efficiently obtain a good matching, we must first transform the empirical reward estimates into a single-peaked matrix -- something standard ETC algorithms cannot do. This required developing a new transformation procedure to project the estimates into a nearby SP matrix. In the known structure regime, a direct application of standard UCB-based optimism would be computationally infeasible, as computing the optimistic matching over all plausible reward matrices is intractable. To overcome this, we proved the existence of a maximal matrix.
>
> **Question 2**: The reviewer asks, "Is there a specific reason for using the term ``preference'' here?" We borrowed this terminology from social choice theory, where single-peakedness originates. In that context, users’ evaluations over alternatives are called preferences, and we adopt the same phrasing for consistency with that literature.

---

> ### Comment · Area_Chair_fUVb · 2025-11-25
>
> Dear Reviewer  jqPn,
>
> Thank you for your time and effort in reviewing the submissions and providing valuable feedback to the authors.
>
> If you haven't already done so, we kindly remind you to review the authors’ rebuttals and engage in the discussion at your earliest convenience. This step ensures efficient communication and helps finalize the process smoothly.
>
> We sincerely appreciate your dedication and collaboration.
>
> Best,
>
> AC

---

> ### Comment · Reviewer_jqPn · 2025-11-25
>
> Thank you for your detailed responses. While the new algorithm is appreciated, my main concern regarding the optimality of the regret bound remains. The paper focuses on worst-case guarantees, yet in the worst case it does not demonstrate improved performance. Although the bound may indeed be tight, the current version does not sufficiently justify this tightness. Therefore, I am maintaining my score.

---

> > ### Author Response · Authors · 2025-11-27
> >
> > Thank you again for your response. Given your continued concern regarding “the optimality of the regret bound,” we would like to clarify the scope of our work.
> >
> > First, **there is no statistical regret gap in our setting**: as we now detail in Section 2.2, an (inefficient) combinatorial algorithm achieves the worst-case statistical optimal regret for both SP and general preferences.
> >
> > Thus, our main contribution lies in **designing efficient algorithms for SP instances**: Exploiting the SP structure required developing new algorithmic tools besides the offline solver that do not arise in standard ETC or UCB methods.
> >
> > Our results indeed point to an **interesting future research**: whether the statistical lower bound can be matched by a polynomial-time algorithm, or whether obtaining such specific sublinear regret is NP-hard; this type of question is rarely asked in the literature.
> >
> > However, **we do not believe this open question detracts from our main contributions**: we are the first to show that SP structure enables efficient learning in an otherwise intractable recommendation setting, and the first to provide polynomial-time algorithms with provable sublinear regret under SP preferences.
> >
> > We would be happy to answer any further questions you may have.

---

### Official Review · Reviewer_6YQV · 2025-10-29

**Soundness:** 3
**Presentation:** 3
**Contribution:** 3
**Rating:** 6
**Confidence:** 3

**Summary:**

This paper studies an online stochastic matching problem where a learner must sequentially match users to arms under a budget constraint, with the goal of maximizing cumulative reward. The general problem is NP-hard, but the authors circumvent this computational barrier by imposing a single-peaked preference structure, a well-known concept from social choice theory where each user's utility is unimodal with respect to a common ordering of the arms. For the offline problem, they develop SP-Matching, an efficient dynamic programming algorithm that finds the optimal budgeted matching. Leveraging this, they propose two online algorithms: EMC, an explore-then-commit algorithm for the challenging case of unknown preference structure, which achieves a regret of $\tilde{O}(UKT^{2/3})$ by extracting an approximate order using PQ trees; and MVM, an efficient UCB-like algorithm for the case of known structure, which achieves a tighter regret of $\tilde{O}(U\sqrt{TK})$ by leveraging a novel maximal matrix within its confidence sets. This paper also gives a regret lower bound analysis for both known and unknown peaks cases.

**Strengths:**

1. The single-peaked preference assumption is grounded in social choice theory and circumvents NP-hardness, enabling efficient algorithms with standard regret bounds instead of weaker α-regret.
2. It provides a complete landscape by offering efficient algorithms for both unknown (EMC) and known (MvM) structure settings, while rigid theoretical analysis are given on both algorithms.
3. The PQ-tree-based order extraction method and the concept of a maximal matrix, which enables optimistic planning while preserving structural constraints, are novel.

**Weaknesses:**

1. While lower bounds are provided, gaps remain between upper and lower bounds in some settings (e.g., for EMC). The paper does not conclusively determine if the attained rates are optimal for polynomial-time algorithms in the unknown structure case. To my knowledge, the ETC-based algorithm is not an order-optimal algorithm in the classical stochastic MAB problem, so it is not surprising that there exists a gap between the upper and lower bounds. Even though the proposed EXTRACT-ORDER algorithm used in EMC provides insight into the construction of an ASP order.
2. The setting that different users can be matched to the same arm simplifies this problem, while a more common setting is the bipartite matching between users and arms.


Minor
1. Title of Section 4.3: written as ETC instead of EMC.

**Questions:**

See weakness.

---

> ### Author Response · Authors · 2025-11-16
>
> We thank the reviewer for their evaluation and feedback. Following the reviewer's first weakness, we intend to both add an algorithm for the unknown structure regime that achieves optimal regret under some assumptions, and to add clarification about our lower bounds that might clarify the nature of the gap. Additionally, following the reviewer's second weakness, we intend to append a dedicated appendix to explore the suggested modeling. We detail these changes below.
>
> **Weakness 1**: The reviewer raises a concern about "The gap between EMC's $O(UKT^{2/3})$ regret and the $\Omega(\sqrt{KT})$ lower bound." We divide our response into two parts:
>
> **New algorithm**: If the entries of the expectation matrix are sufficiently separated, we can devise an efficient and optimal algorithm for the unknown structure regime. Formally, there exists a hybrid algorithm for the unknown structure regime that combines EMC and MvM to obtain regret of approximately
> $$
> O \left(\min ( T^{2/3}, \sqrt{T}+\frac{1}{\Delta^2} ) \right),
> \quad
> \Delta=\min_{u} \min_{i \neq j}|\Theta_{u,i}-\Theta_{u,j}|.
> $$
>
> Importantly, this regret bound strictly improves on that of EMC, and, for $\Delta \geq T^{-1/4}$, it matches the lower bound for this regime. This algorithm follows the spirit of Section 5 (Lines 465–471): first, explore entries of the reward matrix in a round-robin fashion until any two entries in the same row can be reliably compared (within $1/\Delta^2$ rounds by concentration). Then use these estimates to extract the SP order and user peaks and run MvM on the inferred structure, which -- with high probability -- matches the true SP structure. If after at most $T^{2/3}$ rounds the confidence intervals are not disjoint, we switch to the matching and commitment protocol of EMC.
>
> **Clarification about the lower bound and the apparent gap.** As we note in Section 2.2 and Appendix F, an inefficient algorithm can achieve the optimal $O(U\sqrt{TK})$ regret even for non-single-peaked instances. Thus, the challenge in improving regret is *computational*, not *statistical*. Our lower bounds (Theorem 9) are intentionally information-theoretic: they show that SP instances remain statistically non-trivial, since even with the SP structure every algorithm must incur $\Omega(U\sqrt{TK})$ regret. However, these bounds do not incorporate computational constraints and therefore do not match the $O(UKT^{2/3})$ regret achieved by our *efficient* algorithm. Closing this gap would require a fundamentally different type of result, which can be informally stated as “achieving $o(T^{2/3})$ regret in SP instances is NP-hard”. This is a rare and technically challenging type of result in bandit theory. We will clarify in the revision that this mismatch reflects computational, not statistical, limitations.
>
> **Weakness 2**: "The setting that different users can be matched to the same arm simplifies this problem, while a more common setting is the bipartite matching between users and arms."
>
> If the reviewer thinks it will strengthen our paper, we will add a dedicated appendix to analyze this alternative modeling. We detail below the results we obtained for this setting when assuming unit costs ($c_k=1$ for all $k \in [K]$); the general case remains an interesting problem. When assuming unit costs, the offline problem then becomes a budget-constrained bipartite matching problem, and can be solved efficiently using the Hungarian algorithm or a reduction to a min-cost flow problem. This formulation does not require the single-peaked assumption, since the matching problem remains polynomially solvable. Moving to the online setting, one can directly apply an optimistic algorithm that in each round selects the matching maximizing the total UCB estimates. In some sense, when assuming unit costs, this formulation is actually simpler than the one we study as it admits an efficient algorithm with sublinear regret even without assuming any special structure. In contrast, in our model, the offline problem is NP-hard even for unit costs, and structural assumptions are essential to obtain both computational efficiency and sublinear regret. Thanks for the suggestion!

---

> ### Comment · Area_Chair_fUVb · 2025-11-25
>
> Dear Reviewer 6YQV,
>
> Thank you for your time and effort in reviewing the submissions and providing valuable feedback to the authors.
>
> If you haven't already done so, we kindly remind you to review the authors’ rebuttals and engage in the discussion at your earliest convenience. This step ensures efficient communication and helps finalize the process smoothly.
>
> We sincerely appreciate your dedication and collaboration.
>
> Best,
>
> AC

---

### Official Review · Reviewer_486h · 2025-10-30

**Soundness:** 3
**Presentation:** 3
**Contribution:** 2
**Rating:** 4
**Confidence:** 4

**Summary:**

This paper addresses an online stochastic matching problem where a learner sequentially matches users to arms under budget constraints, aiming to maximize cumulative reward over $T$ rounds. The key innovation is leveraging single-peaked (SP) preferences to overcome computational intractability, where user preferences are unimodal with respect to a common arm order. This paper studies 1) offline setting, 2) online setting with known preference structure, and 3) online setting with unknown structure.

**Strengths:**

1. ​​The paper bridges combinatorial bandits and social choice theory by incorporating SP preferences, transforming an NP-hard problem into a tractable one while maintaining standard regret bounds instead of approximate regret.
2. ​​Both offline and online algorithms are designed with theoretical guarantees. The maximal matrix construction for MvM and PQ-tree-based order recovery in EMC demonstrate creative problem-solving.
3. ​​Experimental results demonstrate theoretical regret rates.
4. The budget constraint modeling and SP assumption aligns well with real-world applications like content recommendation.

**Weaknesses:**

My major concern is the gaps in bounds​​ of online setting with unknown structure. The gap between EMC's $O(UKT^{2/3})$ regret and the $\Omega{\sqrt{KT}}$ lower bound suggests potential for improved algorithms or tighter analysis.

**Questions:**

1. Could the SP assumption be relaxed to more general structures while retaining computational efficiency?
2. What is the hardness in designing algorithm attaining $O(\sqrt{T})$ regret for online setting with unknown structure? Is it possible to improve the lower bound by constructing instance with this hardness?

---

> ### Author Response · Authors · 2025-11-16
>
> We thank the reviewer for their evaluation and feedback. Following the reviewer's concern, we intend to both add an algorithm for the unknown structure regime that achieves optimal regret under some assumptions, and to add clarification about our lower bounds that might clarify the nature of the gap. Additionally, your questions made us think of a direct extension of our model to more general preference structures. We address the reviewer's questions below and then move on to address the weaknesses, and hope that our clarifications will contribute to a more positive evaluation of our paper.
>
> **Question 1:"Could the SP assumption be relaxed to more general structures while retaining computational efficiency?"**
> Relaxing the SP assumption is delicate, as large violations might lead to a hard instance of the problem which we cannot hope to solve efficiently. However, given that the violations are small, and an upper bound on their magnitude is known in advance, our current framework can be extended to support this. More specifically, the computational efficiency of our algorithms will remain the same, and an "SP violation term" will be added to the regret bounds. This can be done by our extract-order and projection protocols used in the EMC algorithm -- we can transform the estimated reward matrix to a nearby SP matrix and solve for its optimal matching. The optimality gap can be quantified using the SP violation term. This is an important extension, and we intend to add it to our revision of the paper. Thanks!
>
> **Question 2: "What is the hardness in designing an algorithm attaining $O(\sqrt{T})$ regret for the online setting with unknown structure? Is it possible to improve the lower bound by constructing an instance with this hardness?"**
> As we clarify below, constructing stronger lower bounds is non-trivial, since there already exists an (inefficient) algorithm that attains the optimal statistical bound, so the lower bound must also take into account computational aspects of the algorithms. Regarding the algorithmic design difficulty, we believe the main obstacle lies in the complexity of the action and parameter spaces: the action space is discrete and exponential in $K$, and the parameter space is non-convex and of zero measure among all matrices. This makes it difficult to apply classical techniques from the bandit literature, such as sampling and convex relaxations.
>
> We would also like to address the reviewer's concern about "The gap between EMC's $O(UKT^{2/3})$ regret and the $\Omega(\sqrt{KT})$ lower bound." We divide our response into two parts:
>
> **New algorithm**: If the entries of the expectation matrix are sufficiently separated, we can devise an efficient and optimal algorithm for the unknown structure regime. Formally, there exists a hybrid algorithm for the unknown structure regime that combines EMC and MvM to obtain regret of approximately
> $$
> O \left(\min ( T^{2/3}, \sqrt{T}+\frac{1}{\Delta^2})\right),
> \quad
> \Delta=\min_{u} \min_{i \neq j}|\Theta_{u,i}-\Theta_{u,j}|.
> $$
> Importantly, this regret bound strictly improves on that of EMC, and, for $\Delta \geq T^{-1/4}$, it matches the lower bound for this regime. This algorithm follows the spirit of Section 5 (Lines 465–471): first, explore entries of the reward matrix in a round-robin fashion until any two entries in the same row can be reliably compared (within $1/\Delta^2$ rounds by concentration). Then use these estimates to extract the SP order and user peaks and run MvM on the inferred structure, which -- with high probability -- matches the true SP structure. If after at most $T^{2/3}$ rounds the confidence intervals are not disjoint, we switch to the matching and commitment protocol of EMC.
>
> **Clarification about the lower bound and the apparent gap.** As we note in Section 2.2 and Appendix F, an inefficient algorithm can achieve the optimal $O(U\sqrt{TK})$ regret even for non-single-peaked instances. Thus, the challenge in improving regret is *computational*, not *statistical*. Our lower bounds (Theorem 9) are intentionally information-theoretic: they show that SP instances remain statistically non-trivial, since even with the SP structure every algorithm must incur $\Omega(U\sqrt{TK})$ regret. However, these bounds do not incorporate computational constraints and therefore do not match the $O(UKT^{2/3})$ regret achieved by our *efficient* algorithm. Closing this gap would require a fundamentally different type of result, which can be informally stated as “achieving $o(T^{2/3})$ regret in SP instances is NP-hard”. This is a rare and technically challenging type of result in bandit theory. We will clarify in the revision that this mismatch reflects computational, not statistical, limitations.

---

> > ### Comment · Reviewer_486h · 2025-11-27
> >
> > Thanks for your response. It has addressed some of my concerns about the lower bound analysis. I will increase my score.

---

> ### Comment · Area_Chair_fUVb · 2025-11-25
>
> Dear Reviewer 486h,
>
> Thank you for your time and effort in reviewing the submissions and providing valuable feedback to the authors.
>
> If you haven't already done so, we kindly remind you to review the authors’ rebuttals and engage in the discussion at your earliest convenience. This step ensures efficient communication and helps finalize the process smoothly.
>
> We sincerely appreciate your dedication and collaboration.
>
> Best,
>
> AC

---

### Official Review · Reviewer_hCLu · 2025-11-01

**Soundness:** 3
**Presentation:** 3
**Contribution:** 3
**Rating:** 6
**Confidence:** 3

**Summary:**

This paper studies a budget-constrained matching problem. The authors first point out that, under general preference structures, the problem is NP-hard even in the offline setting. To overcome this computational barrier, the paper introduces the assumption of single-peaked (SP) preferences, which theoretically ensures tractability. For the offline case, the authors propose a dynamic programming algorithm, SP-MATCHING, which computes the optimal matching in polynomial time under a known SP order. This algorithm provides the theoretical and computational foundation for the subsequent online methods. In the online setting, the paper considers two cases:

1.	When the SP structure is known, the authors propose an optimistic algorithm MvM (Match-via-Maximal) based on maximal matrices, which uses SP-MATCHING each round to compute the optimal matching and achieves sublinear regret $O(\sqrt{T})$.
2. When the SP structure is unknown, the authors propose an explore-then-commit (EMC) method that first learns an approximate preference order from data collected in a round-robin manner via an EXTRACT-ORDER process based on PQ trees, and then executes this policy based on the approximate preference order, achieving regret $O(T^{2/3})$.
The paper further establishes corresponding regret lower bounds for the known and unknown peak settings. Experiments on synthetic data validate the theoretical results.

**Strengths:**

1. The paper introduces the single-peaked (SP) structural assumption for budget-constrained matching, turning an otherwise NP-hard problem into one solvable in polynomial time. This may offer a principled blueprint for simplifying other computationally hard online learning settings.
2. The algorithmic design is well structured: SP-MATCHING yields a polynomial-time optimal solution offline; the online setting presents MvM (known SP) and EMC (unknown SP), both built on SP-MATCHING. The theory is tight and carefully argued, with detailed proofs of regret bounds and supporting lemmas in the appendix. The results are complete with both upper bounds and lower bounds provided.

**Weaknesses:**

1.	Strong assumptions.
The paper relies on the single-peaked (SP) preference assumption, which may be overly idealized in complex real-world environments. Although the paper states that this preference structure is common in various domains, such as recommendation systems (Line 049-052), no reference is provided for justification. The additional requirement of known SP order and user peaks for the MvM algorithm is unrealistic, which limits its applicability.

2. Insufficient experiments.

This paper introduces a new matching setting with a budget constraint, which appears to have not been studied before. The experiments are also restricted to synthetic data. No real-world applications/datasets have been tested. Moreover, the paper lacks comparisons with general combinatorial bandit baselines, particularly those without SP assumptions but with α-regret guarantees, which would better contextualize the claimed advantages.

**Questions:**

1. For the UCB-based algorithm, the authors claim that solving the optimization problem requires a known SP order. How about using the estimated order from the estimated means and UCB/LCBs? Specifically, though the total order is initially unknown,  the algorithm can construct a partial ordering based on estimations. Can the algorithms solve some optimization problem corresponding to partial ordering to avoid the strong assumption and get an $O(\sqrt{T})$ regret? The authors can discuss this point.
2. Writing suggestion (optional): When two algorithms are available, usually the paper first presents an algorithm that requires strong assumptions to convey some intuition and then presents a more general one.

---

> ### Author Response · Authors · 2025-11-16
>
> We thank the reviewer for their thoughtful evaluation. Notably, following the reviewers first weakness, we plan to add to the revision of the paper an extension of our results to a more relaxed preferences structure. We answer below the reviewer's questions and then move on to address their weaknesses.
>
> **Question 1: "For the UCB-based algorithm... how about using the estimated order from the estimated means and UCB/LCBs?"**
> We explored this direction, but our attempts revealed some difficulties. We see two possible implementations:
>
> 1. **Dynamic ASP ordering:** We can maintain an upper-confidence matrix throughout the interaction; at each round $t$, extract an ASP order, project the matrix to be SP, and then run SP-Matching. The main limitation is that some entries might remain under-explored, which hurt the approximation quality we get from the SP projection.
>
> 2. **Enumerating plausible SP orders:** We can maintain a PQ-tree of all orders and peaks consistent with current confidence intervals, and select the optimistic matching among them. While it might lead to optimal regret, this approach is computationally infeasible: the number of consistent SP orders can be exponential in $K$ and will only grow when considering all the matrices in the confidence set, making per-round optimization over all of those orders infeasible.
>
> We would be happy to explore the reviewer’s intended variant if it differs from these two interpretations.
>
> **Question 2: Writing suggestion**
> We appreciate the reviewer's suggestion. Indeed, we have reflected on this idea before submitting the paper. We eventually decided to go with the current order because EMC was easier to explain and it helped clarify why directly relying on the offline solver is insufficient in the online setting. However, if the reviewer still thinks the paper would benefit from this change, we will consider it again.
>
> We would also like to address the reviewer's raised weaknesses.
>
> **Weakness 1: "The paper relies on the single-peaked preference assumption, which may be overly idealized in complex real-world environments"**
> Relaxing the SP assumption is delicate, as large violations might lead to a hard instance of the problem which we cannot hope to solve efficiently. However, following the reviewer's suggestion, we were able to extend our current framework to support more general structures, given that the SP violations are small and an upper bound on their magnitude is known in advance. More specifically, the computational efficiency of our algorithms will remain the same, and an "SP violation term" will be added to the regret bounds. This can be done by our extract-order and projection protocols used in the EMC algorithm -- we can transform the estimated reward matrix to a nearby SP matrix and solve for its optimal matching, and the optimality gap can be quantified using the SP violation term. This is an important extension, and we intend to add it to our revision of the paper. Thanks!
>
> **Weakness 2: "The additional requirement of known SP order and user peaks for the MvM algorithm is unrealistic, which limits its applicability."**
> We agree that these assumptions are stronger and limit generality. Nonetheless, we believe they are realistic in certain domains. For example, in some settings--such as political podcasts--the SP order can often be inferred from contextual knowledge (e.g., a known ideological spectrum), and user peaks can be identified from user profiles or past behavior. Importantly, identifying only the *index* of the peak, rather than its cardinal value, is sufficient. Moreover, as discussed in Lines 465–471, the assumption of known peaks can be partially relaxed under a separation condition.
>
> **Weakness 3: "The paper lacks comparisons with general combinatorial bandit baselines."**
> In Appendix F, we already compare the "traditional" combinatorial bandit methods adapted to our setting with our new algorithms. Specifically, we show (i) an inefficient algorithm that achieves optimal regret and (ii) an efficient algorithm achieving sublinear $\alpha$-regret. In the revision, we plan to clarify the role of our inefficient algorithm and to elaborate on how $\alpha$-regret approaches could be instantiated in our setting. Thanks!

---

> ### Comment · Area_Chair_fUVb · 2025-11-25
>
> Dear Reviewer hCLu,
>
> Thank you for your time and effort in reviewing the submissions and providing valuable feedback to the authors.
>
> If you haven't already done so, we kindly remind you to review the authors’ rebuttals and engage in the discussion at your earliest convenience. This step ensures efficient communication and helps finalize the process smoothly.
>
> We sincerely appreciate your dedication and collaboration.
>
> Best,
>
> AC

---

### Author Response · Authors · 2025-11-16

Dear Reviewers,

Thank you for your time and effort in reviewing our paper. Following the reviewers’ constructive comments, we intend to use the additional page and additional appendices to add new algorithms, clarify a subtle point in the paper, and extend our results. We estimate it will take us about 7 days to incorporate these changes, after which we will upload a revised version of the paper. We list these changes below:

1. **Including the optimal, inefficient algorithm in the body of the paper.** We mentioned in Section 2.2 about the existence of an optimal, inefficient algorithm that can be used also under the SP assumption. We formally presented this algorithm in Appendix F. However, the reviewers' responses hint that the algorithm's concise presentation in the body of the paper is not sufficient, and we intend to elaborate on this algorithm and its guarantees.
2. **Putting the lower bound in the right context.** Following the previous point, we already have an optimal inefficient algorithm that achieve our lower bounds. However, devising lower bounds for efficient algorithms is a fundamentally different problem, on which we intend to elaborate.
3.  **An efficient and optimal algorithm for separated instances.** Under some instance assumptions, we develop an efficient and optimal algorithm for the unknown structure regime.
4. **Solving almost-SP instances.** Following the reviewers' suggestions, we come up with a natural extension of our results to instances which are not exactly single-peaked.

We detail about these changes below:

**Including the optimal, inefficient algorithm in the body of the paper.**

When the underlying expectation matrix is not SP, no *efficient* algorithm can achieve sublinear regret. However, as explained in Appendix F, there exists an *inefficient* algorithm that achieves a regret of $O(U\sqrt{TK})$. We intend to write this formally in the body of the paper. This algorithm can be viewed an inefficient variant of MvM. Like MvM, it maintains a UCB index for each (user, item) pair and, in each round, selects the optimal matching with respect to this index matrix. Unlike MvM, it is inefficient because the index matrix lacks exploitable structure.

**Putting the lower bound in the right context.**

As we note in Section 2.2 and Appendix F, an inefficient algorithm can achieve the optimal $O(U\sqrt{TK})$ regret even for non-single-peaked instances. Thus, the challenge in improving regret is *computational*, not *statistical*. Our lower bounds (Theorem 9) are intentionally information-theoretic: they show that SP instances remain statistically non-trivial, since even with the SP structure every algorithm must incur $\Omega(U\sqrt{TK})$ regret. However, these bounds do not incorporate computational constraints and therefore do not match the $O(UKT^{2/3})$ regret achieved by our *efficient* algorithm. Closing this gap would require a fundamentally different type of result, which can be informally stated as “achieving $o(T^{2/3})$ regret in SP instances is NP-hard”. This is a rare and technically challenging type of result in bandit theory. We will clarify in the revision that this mismatch reflects computational, not statistical, limitations.

**An efficient and optimal algorithm for separated instances.**:

If the entries of the expectation matrix are sufficiently separated, we can devise an efficient and optimal algorithm for the unknown structure regime. Formally, there exists a hybrid algorithm for the unknown structure regime that combines EMC and MvM to obtain regret of approximately
$$
O \left(\min ( T^{2/3}, \sqrt{T}+\frac{1}{\Delta^2} ) \right),
\quad
\Delta=\min_{u} \min_{i \neq j}|\Theta_{u,i}-\Theta_{u,j}|.
$$
Importantly, this bound strictly improves on that of EMC, and for $\Delta \geq T^{-1/4}$, it matches the lower bound for this regime. For more details on this algorithm, please see our elaborated responses to reviewers 486h, 6YQV and jqPn.

**Solving almost-SP instances**:
We can extend our algorithmic framework to handle instances which violate the SP assumption by a small amount. This can be done by our extract-order and projection protocols used in the EMC algorithm -- we can transform the estimated reward matrix to a nearby SP matrix and solve for its optimal matching, and the optimality gap can be quantified using an SP violation term.

---

> ### Author Response · Authors · 2025-11-23
>
> Dear reviewers,
>
> We would like to update you that we have uploaded a revised version of the paper, incorporating all the changes described in our responses. Below, we summarize the main modifications together with their locations in the revision.
>
> 1. **"Including the optimal, inefficient algorithm in the body of the paper."** - *lines 182-188.*
>
> We added a corollary stating the existence of an inefficient algorithm that achieves optimal regret (for both the general and SP structures) and included an example for such an algorithm.
>
> 2. **"Putting the lower bound in the right context."** - *lines 238-254, 528-534.*  (Following the feedback from reviewers 486h, 6YQV, jqPn)
>
> We moved the lower bounds section to the end of Section 2, immediately after introducing the SP structure. This better highlights its intended role: showing that while SP preferences remove the computational barrier present in the general case, they do not make the problem statistically easier. We also explain that the inefficient algorithm is optimal for both SP and general instances. Additionally, using the extra page, we brought the discussion section into the main text. There, we describe the remaining room for improvement between our algorithmic guarantees and our lower bounds: the statistical bounds match, and the gap is purely computational. The open question of whether one can design an efficient algorithm achieving the statistical lower bound is now stated explicitly.
>
> 3. **"An efficient and optimal algorithm for separated instances."** - *lines 480-495.*  (Following the feedback from reviewers 486h, 6YQV, jqPn)
>
> We added a subsection describing Sep-MvM, a hybrid algorithm for the unknown-structure regime. Sep-MvM strictly improves upon EMC and is optimal when the instance is sufficiently separated (Proposition 12).
>
> 4. **"Solving almost-SP instances."** - *lines 497-515.*  (Following the feedback from reviewers hCLu, 486h)
>
> We have introduced the NSP-EMC algorithm, which extends the EMC algorithm to accommodate non-single-peaked matrices. NSP-EMC maintains essentially the same runtime and regret guarantees, up to an additional term depending on the instance’s distance from single-peakedness (Proposition 13). We provided a brief proof sketch of Proposition 13 to exemplify its approach.
>
> Thank you again for your time and insightful feedback. We look forward to your further comments on this revised submission.

---

### Meta-Review · Area_Chair_VqB6 · 2026-01-07

**Summary:**

This paper studies a budget-constrained stochastic matching problem where users are repeatedly matched to arms under a knapsack-style budget. In the absence of structure, even the offline optimization problem is NP-hard to approximate beyond a constant factor, which makes standard online learning approaches computationally infeasible. The paper’s main contribution is to show that assuming single-peaked (SP) preferences removes this computational barrier while preserving the statistical difficulty of the problem. Under SP structure, the authors provide a polynomial-time dynamic programming algorithm (SP-MATCHING) for the offline problem, and then build online algorithms on top of it.

For the online setting, the paper considers two regimes. When the SP order and user peaks are known, the proposed UCB-style algorithm (MVM) leverages a “maximal matrix” construction to achieve regret on the order of $\tilde O(U\sqrt{TK})$. When the structure is unknown, the explore-then-commit (EMC) algorithm first estimates rewards, recovers an approximate SP order using a PQ-tree-based procedure, projects the estimates onto a nearby SP matrix, and then commits to the resulting matching, yielding regret $\tilde O(UKT^{2/3})$. The revised version clarifies the lower-bound landscape, adds extensions for separated instances and near-SP matrices, and explicitly frames the remaining gap as a computational (rather than statistical) open question.

**Reviewer Concerns:**

A central concern across reviewers is the gap between the regret upper bound for the unknown-structure case and the statistical lower bound. While the authors convincingly argue that the $\Omega(U\sqrt{TK})$ lower bound is information-theoretic and achievable by inefficient algorithms, the paper does not establish whether the $\tilde O(UKT^{2/3})$ rate of EMC is optimal among polynomial-time algorithms. The additional discussion and the new “separated instance” result help contextualize this issue, but the worst-case gap for efficient algorithms appears to remain.

Reviewers also pointed out that the assumptions underlying the strongest results are restrictive. In particular, MVM assumes the SP order and user peaks are known, which may limit applicability in practice. The paper partially addresses this by motivating scenarios where such information could be available and by adding robustness results for near-SP instances, but the realism of these assumptions and their prevalence in real data are not empirically validated.

Finally, some reviewers were concerned that the experimental evaluation is limited. There are no comparisons with standard combinatorial bandit baselines.

**Reviewer Scores:**

One reviewer (486h) explicitly stated that the clarification of the lower-bound context addressed their main concern and that they would increase their score. The AC would also agree with such an update.

Another reviewer (jqPn) explicitly maintained their score, citing the unresolved optimality gap in the unknown-structure regime.

The remaining reviewers, who were already marginally positive, did not raise new notable issues after the revision and would likely have kept their scores unchanged.

Overall, this points to a mixed but expected to have positive shift.

---

### Decision · Program_Chairs · 2026-01-26

Accept (Poster)